# Variable Complexity Weighted-Tempered Gibbs Samplers for Bayesian Variable Selection

## Abstract

A subset weighted-tempered Gibbs Sampler (subset-wTGS) has been recently introduced by Jankowiak to reduce the computation complexity per MCMC iteration in high-dimensional applications where the exact calculation of the posterior inclusion probabilities (PIP) is not essential. However, the Rao-Backwellized estimator associated with this sampler has a very high variance as the ratio between the signal dimension, $P$, and the number of conditional PIP estimations is large. In this paper, we design a new subset-wTGS where the expected number of computations of conditional PIPs per MCMC iteration can be much smaller than $P$. Different from the subset-wTGS and wTGS, our sampler has a variable complexity per MCMC iteration. We provide an upper bound on the variance of an associated Rao-Blackwellized estimator for this sampler at a finite number of iterations, $T$, and show that the variance is $O\left(\left(\frac{P}{S}\right)^2 \frac{\log T}{T}\right)$ for any given dataset where $S$ is the expected number of conditional PIP computations per MCMC iteration.

## 1 Introduction

Markov chain Monte Carlo (MCMC) methods comprise a class of algorithms for sampling from a known function. MCMC methods are primarily used for calculating numerical approximations of multi-dimensional integrals, for example in Bayesian statistics, computational physics (Kasim et al., 2019), computational biology, (Gupta & Rawlings, 2014), and linear models (Truong, 2022). Monte Carlo algorithms have been very popular over the last decade (Hesterberg, 2002; Robert & Casella, 2005). Many practical problems in statistical signal processing, machine learning and statistics, demand fast and accurate procedures for drawing samples from probability distributions that exhibit arbitrary, non-standard forms (Andrieu et al., 2004; Fitzgerald, 2001; Read et al., 2012). One of the most popular Monte Carlo methods are the families of Markov chain Monte Carlo (MCMC) algorithms (Andrieu et al., 2004; Robert & Casella, 2005) and particle filters (Bugallo et al., 2007). Particle filters, or sequential Monte Carlo methods, are a set of Monte Carlo algorithms used to find approximate solutions for filtering problems for nonlinear state-space systems, such as signal processing and Bayesian statistical inference (Wills & Schön, 2023). The MCMC techniques generate a Markov chain with a pre-established target probability density function as invariant density (Liang et al., 2010).

Gibbs sampler (GS) is a Markov chain Monte Carlo (MCMC) algorithm for obtaining a sequence of observations from a specific multivariate probability distribution. This sequence can be used to approximate the joint distribution, the marginal distribution of one of the variables, or some subset of the variables. It can be also used to compute the expected value (integral) of one of the variables (Bishop, 2006; Bolstad, 2010). GS is applicable when the joint distribution is not known explicitly or is difficult to sample from directly, but the conditional distribution of each variable is known and is easy (or at least, easier) to sample from. The GS algorithm generates an instance from the distribution of each variable in turn, conditional on the current values of the other variables. It can be shown that the sequence of samples constitutes a Markov chain, and the stationary distribution of that Markov chain is just the sought-after joint distribution.

GS is commonly used as a means of statistical inference, especially Bayesian inference. However, pure Markov chain based schemes (i.e., ones which simulate from precisely the right target distribution with no need for

subsequent importance sampling correction) have been far more successful. This is because MCMC methods are usually much more scalable to high-dimensional situations, whereas importance sampling weight variances tend to grow (often exponentially) with dimension. (Zanella & Roberts, 2019) proposed a natural way to combine the best of MCMC and importance sampling in a way that is robust in high-dimensional contexts and ameliorates the slow mixing which plagues many Markov chain based schemes. The proposed scheme is called Tempered Gibbs Sampler (TGS), involving component-wise updating rule like Gibbs Sampling (GS), with improved mixing properties and associated importance weights which remain stable as dimension increases. Through an appropriately designed tempering mechanism, TGS circumvents the main limitations of standard GS, such as the slow mixing introduced by strong posterior correlations. It also avoids the requirement to visit all coordinates sequentially, instead iteratively making state-informed decisions as to which coordinate should be next updated.

TGS has been applied to Bayesian Variable Selection (BVS) problem, observing multiple orders of magnitude improvements compared to alternative Monte Carlo schemes (Zanella & Roberts, 2019). Since TGS updates each coordinate with the same frequency, in a BVS context, this may be inefficient as the resulting sampler would spend most iterations updating variables that have low or negligible posterior inclusion probability, especially when the number of covariates, $P$, gets large. A better solution, called weighted Tempered Gibbs Sampling (wTGS) (Zanella & Roberts, 2019), updates more often components with a larger inclusion probability, thus having a more focused computational effort. However, despite the intuitive appeal of this approach to BVS problem, approximating the resulting posterior distribution can be computationally challenging. A principal reason for this is the astronomical size of the model space whenever there more than a few dozen covariates. To scale the high-dimensional regime, (Jankowiak, 2023) has recently introduced an efficient MCMC scheme whose cost per iteration can be significantly reduced compared to wTGS. The main idea is to introduce an auxiliary variable $\mathcal{S} \subset \{1, 2, \cdots, P\}$ that controls which conditional posterior inclusion probabilites (PIPs) are computed in a given MCMC iteration. By choosing the size $S$ of $\mathcal{S}$ to be much less than $P$, we can reduce the computational complexity significantly. However, this scheme contains some weaknesses such as the Rao-Blackwellized estimator associated with this sampler has a very high variance when $P/S$ is large and the number of MCMC iterations, $T$, is small. In addition, generating the auxiliary random set which is uniformly distributed over $\binom{P}{S}$ subsets in the subset wTGS algorithm (Jankowiak, 2023) requires very long running time.

In this paper, we design a new subset wTGS called variable complexity wTGS (VC-wTGS) and apply this algorithm to BVS in the linear regression model. More specifically, we consider the linear regression $Y = X\beta + Z$ where $\beta = (\beta_0, \beta_1, \ldots, \beta_{P-1})^T$ is controlled by an inclusion vector $(\gamma_0, \gamma_1, \cdots, \gamma_{P-1})$. We design a Rao-Blackwellized estimator associated with VC-wTGS for *posterior inclusion probabilities* or PIPs, where $\texttt{PIP}(i) := p(\gamma_i = 1 | \mathcal{D}) \in [0, 1]$, and $\mathcal{D} = \{X, Y\}$ is the observed dataset. Experiments show that our scheme converges to PIPs very fast for simulated datasets and that the variance of the Rao-Blackwellized estimator can be much smaller than the subset wTGS (Jankowiak, 2023) when $P/S$ is very high for MNIST dataset. More specifically, our contributions include:

- We propose a new subset wTGS, called VC-wTGS, where the expected number of conditional PIP computations per MCMC can be much smaller than the signal dimension.

- We analyse the variance of an associated Rao-Blackwellized estimator at each finite number of MCMC iterations. We show that this variance is $O\left(\frac{\log T}{T}\left(\frac{P}{S}\right)^2\right)$ for any given dataset.

- We provide some experiments on a simulated dataset (multivariate Gaussian dataset) and the real dataset (MNIST). Experiments show that our estimator can have a better variance than the subset wTGS-based estimator (Jankowiak, 2023) at high $P/S$ for the same number of MCMC iterations $T$.

Although we limit our application to the linear regression model for the simplicity of computations of the conditional PIPs in experiments, our subset wTGS can be applied to other BVS models. However, we need to change the method to estimate the conditional PIPs for each model. See (148) and Appendix E for the method that is used to estimate the conditional PIPs for the linear regression model.

## 2 Preliminaries

### 2.1 Mathematical Backgrounds

Let a Markov chain $\{X_n\}_{n=1}^{\infty}$ on a state space $\mathcal{S}$ with transition kernel $Q(x, dy)$ and the initial state $X_1 \sim \nu$, where $\mathcal{S}$ is a Polish space in $\mathbb{R}$. In this paper, we consider the Markov chains which are irreducible and positive-recurrent, so the existence of a stationary distribution $\pi$ is guaranteed. An irreducible and recurrent Markov chain on an infinite state-space is called Harris chain (Tuominen & Tweedie, 1979). A Markov chain is called *reversible* if the following detailed balance condition is satisfied:

$$\pi(dx)Q(x, dy) = \pi(dy)Q(y, dx), \qquad \forall x, y \in \mathcal{S}. \tag{1}$$

Define

$$d(t) := \sup_{x \in \mathcal{S}} d_{\mathrm{TV}}(Q^t(x, \cdot), \pi) \tag{2}$$

$$t_{\mathrm{mix}}(\varepsilon) := \min\{t : d(t) \le \varepsilon\}, \tag{3}$$

and

$$\tau_{\min} := \inf_{0 \le \varepsilon \le 1} t_{\mathrm{mix}}(\varepsilon) \left( \frac{2 - \varepsilon}{1 - \varepsilon} \right)^2, \quad t_{\mathrm{mix}} := t_{\mathrm{mix}}(1/4). \tag{4}$$

Let $L_2(\pi)$ be the Hilbert space of complex valued measurable functions on $\mathcal{S}$ that are square integrable w.r.t. $\pi$. We endow $L_2(\pi)$ with inner product $\langle f, g \rangle := \int fg^* d\pi$, and norm $\|f\|_{2,\pi} := \langle f, f \rangle_\pi^{1/2}$. Let $E_\pi$ be the associated averaging operator defined by $(E_\pi)(x, y) = \pi(y), \forall x, y \in \mathcal{S}$, and

$$\lambda = \|Q - E_\pi\|_{L_2(\pi) \to L_2(\pi)}, \tag{5}$$

where $\|B\|_{L_2(\pi) \to L_2(\pi)} = \max_{v : \|v\|_{2,\pi}=1} \|Bv\|_{2,\pi}$. $Q$ can be viewed as a linear operator on $L_2(\pi)$, denoted by $\mathbf{Q}$, defined as $(\mathbf{Q}f)(x) := \mathbb{E}_{Q(x,\cdot)}(f)$, and the reversibility is equivalent to the self-adjointness of $\mathbf{Q}$. The operator $\mathbf{Q}$ acts on measures on the left, creating a measure $\mu\mathbf{Q}$, that is, for every measurable subset $A$ of $\mathcal{S}$, $\mu\mathbf{Q}(A) := \int_{x \in \mathcal{S}} Q(x, A)\mu(dx)$. For a Markov chain with stationary distribution $\pi$, we define the *spectrum* of the chain as

$$S_2 := \{\xi \in \mathbb{C} : (\xi\mathbf{I} - \mathbf{Q}) \text{ is not invertible on } L_2(\pi)\}. \tag{6}$$

It is known that $\lambda = 1 - \gamma^*$ (Paulin, 2015), where

$$\gamma^* := \begin{cases} 1 - \sup\{|\xi| : \xi \in \mathcal{S}_2, \xi \ne 1\}, \\ \qquad \text{if eigenvalue 1 has multiplicity 1,} \\ 0, \qquad \text{otherwise} \end{cases}$$

is the *the absolute spectral gap* of the Markov chain. The absolute spectral gap can be bounded by the mixing time $t_{\mathrm{mix}}$ of the Markov chain by the following expression:

$$\left( \frac{1}{\gamma^*} - 1 \right) \log 2 \le t_{\mathrm{mix}} \le \frac{\log(4/\pi_*)}{\gamma_*}, \tag{7}$$

where $\pi_* = \min_{x \in \mathcal{S}} \pi_x$ is the *minimum stationary probability*, which is positive if $Q^k > 0$ (entry-wise positive) for some $k \ge 1$. See (Wolfer & Kontorovich, 2019) for more detailed discussions. In (Combes & Touati, 2019; Wolfer & Kontorovich, 2019), the authors provided algorithms to estimate $t_{\mathrm{mix}}$ and $\gamma^*$ from a single trajectory.

Let $\mathcal{M}(S)$ be a measurable space on $\mathcal{S}$ and define

$$\mathcal{M}_2 := \left\{ \nu \text{ defined on } \mathcal{M}(\mathcal{S}) : \nu << \pi, \left\| \frac{d\nu}{d\pi} \right\|_2 < \infty \right\}, \tag{8}$$

where $\| \cdot \|_2$ is the standard $L_2$ norm in the Hilbert space of complex valued measurable functions on $\mathcal{S}$.

## 2.2 Problem Set-up

Consider the linear regression $Y = X\beta + Z \in \mathbb{R}^N$ where $\beta = (\beta_0, \beta_1, \ldots, \beta_{P-1})^T$, $Z = (Z_0, Z_1, \ldots, Z_{P-1})^T$, and $X \in \mathbb{R}^{N \times P}$ which is a designed matrix. Denote $\gamma$ by the vector $(\gamma_0, \gamma_1, \cdots, \gamma_{P-1})$ where each $\gamma_i \in \{0, 1\}$ controls whether the coefficient $\beta_i$ and the $i$-th covariate are included ($\gamma_i = 1$) or excluded ($\gamma_i = 0$) from the model. Let $\beta_\gamma$ be the restriction of $\beta$ to the coordinates in $\gamma$ and $|\gamma| \in \{0, 1, 2, \cdots, P\}$ be the total number of included covariates. In addition, the following are assumed:

- inclusion variables: $\gamma_i \sim \mathtt{Bern}(h)$

- noise variance: $\sigma_\gamma^2 \in \mathtt{InvGamma}\left(\frac{1}{2}\nu_0, \frac{1}{2}\nu_0\lambda_0\right)$

- coefficients: $\beta_\gamma \sim \mathcal{N}(0, \sigma_\gamma^2 \tau^{-1} \mathbf{I}_{|\gamma|})$

- noise distributions: $Z_i \sim \mathcal{N}(0, \sigma_\gamma^2)$

for all $i = 0, 1, \cdots, P-1$. The hyperparameter $h \in (0, 1)$ controls the overall level of sparsity; in particular $hP$ is the expected number of covariates included a priori. The $|\gamma|$ coefficients $\beta_\gamma \in \mathbb{R}^{|\gamma|}$ are governed by the standard Gaussian prior with precision proportional to $\tau > 0$.

An attractive feature of the model is that it explicitly reasons about variable inclusion and allows us to define *posterior inclusion probabilities* or PIPs, where

$$\mathtt{PIP}(i) := p(\gamma_i = 1 | \mathcal{D}) \in [0, 1], \tag{9}$$

and $\mathcal{D} = \{X, Y\}$ is the observed dataset.

# 3 Main Results

## 3.1 Introduction to Subset wTGS

In this subsection, we review the subset wTGS which was proposed by (Jankowiak, 2023). Let $\mathcal{P} = \{1, 2, \cdots, P\}$ and $\mathcal{P}_S$ be the set of all subsets of cardinality $S$ of $\mathcal{P}$. Consider the sample space $\mathcal{P} \times \{0, 1\}^P \times \mathcal{P}_S$ and define the following (unnormalized) target distribution on this sample space:

$$f(\gamma, i, \mathcal{S}) := p(\gamma | \mathcal{D}) \frac{\frac{1}{2}\eta(\gamma_{-i})}{p(\gamma_i | \gamma_{-i}, \mathcal{D})} \mathcal{U}(\mathcal{S} | i, \mathcal{A}). \tag{10}$$

Here, $\mathcal{S}$ ranges over all the subsets of $\{1, 2, \cdots, P\}$ of some size $S \in \{0, 1, \cdots, P\}$ that also contain a fixed 'anchor' set $\mathcal{A} \subset \{1, 2, \cdots, P\}$ of size $A < S$, and $\eta(\cdot)$ is some weighting function. Moreover, $\mathcal{U}(\mathcal{S} | i, \mathcal{A})$ is the uniform distribution over the all size $S$ subsets of $\{1, 2, \cdots, P\}$ that contain both $i$ and $\mathcal{A}$.

In practice, the set $\mathcal{A}$ can be chosen during burn-in. Subset wTGS proceeds by defining a sampling scheme for the target distribution (10) that utilizes Gibbs updates w.r.t. $i$ and $\mathcal{S}$ and Metropolized-Gibbs update w.r.t. $\gamma_i$.

- **$i$-updates:** Marginalizing $i$ from (10) yields

$$f(\gamma, \mathcal{S}) = p(\gamma | \mathcal{D}) \phi(\gamma, \mathcal{S}) \tag{11}$$

where we define

$$\phi(\gamma, \mathcal{S}) := \sum_{i \in \mathcal{S}} \frac{\frac{1}{2}\eta(\gamma_{-i})}{p(\gamma_i | \gamma_{-i}, \mathcal{D})} \mathcal{U}(\mathcal{S} | i, \mathcal{A}) \tag{12}$$

and have leveraged that $\mathcal{U}(\mathcal{S} | i, \mathcal{A}) = 0$ if $i \notin \mathcal{S}$. Crucially, computing $\phi(\gamma, \mathcal{S})$ is $\Theta(S)$ instead of $\Theta(P)$. We can do Gibbs updates w.r.t. $i$ using the distribution

$$f(i | \gamma, \mathcal{S}) \sim \frac{\eta(\gamma_{-i})}{p(\gamma_i | \gamma_{-i}, \mathcal{D})} \mathcal{U}(\mathcal{S} | i, \mathcal{A}). \tag{13}$$

- $\gamma$-**updates:** Just as for *wTGS* we utilized Metropolized -Gibbs updates w.r.t. $\gamma_i$ that result in deterministic flips $\gamma_i \to 1 - \gamma_i$. Likewise the marginal $f(i)$ is proportional to $\texttt{PIP}(i) + \frac{\varepsilon}{P}$ so that the sampler focuses computational efforts on large PIP covariates (Jankowiak, 2023).

- $\mathcal{S}$-**updates:** $\mathcal{S}$ is updated with Gibbs moves, $\mathcal{S} \sim \mathcal{U}(\cdot|i, \mathcal{A})$. For the full algorithm, see the Algorithm 1.

---

**Algorithm 1** The Subset $\mathcal{S}$-wTGS Algorithm

---

**Input:** Dataset $\mathcal{D} = \{X, Y\}$ with $P$ covariates; prior inclusion probability $h$; prior precision $\tau$; subset size $S$; anchor set size $A$; total number of MCMC iterations $T$; number of burn-in iteration $T_{\text{burn}}$.

**Output:** Approximate weighted posterior samples $\{\rho^{(t)}, \gamma^{(t)}\}_{t=T_{\text{burn}}+1}^{T}$

**Initializations:** $\gamma^{(0)} = (1, 1, \cdots, 1)$, and choose $\mathcal{A}$ be the $A$ covariate with exhibiting the largest correlations with $Y$. Choose $i^{(0)}$ randomly from $\{1, 2, \cdots, P\}$ and $\mathcal{S}^{(0)} \sim \mathcal{U}(\cdot|i^{(0)}, \mathcal{A})$.

**for** $t = 1, 2, \cdots, T$ **do**

Estimate $S$ conditional PIPs $p(\gamma_j^{(t-1)}|\gamma_{-j}^{(t-1)}, \mathcal{D})$ for all $j \in \mathcal{S}^{(t-1)}$

$\phi(\gamma^{(t-1)}, \mathcal{S}^{(t-1)}) \leftarrow \sum_{j \in \mathcal{S}^{(t-1)}} \frac{\frac{1}{2}\eta(\gamma_{-j}^{(t-1)})}{p(\gamma_j^{(t-1)}|\gamma_{-j}^{(t-1)}, \mathcal{D})}$

Estimate $f(j|\gamma^{(t-1)}) \leftarrow \phi^{-1}(\gamma^{(t-1)}, \mathcal{S}^{(t-1)}) \frac{\frac{1}{2}\eta(\gamma_{-j}^{(t-1)})}{p(\gamma_j^{(t-1)}|\gamma_{-j}^{(t-1)}, \mathcal{D})}$ for all $j \in [P]$.

Sample $i^{(t)} \sim f(\cdot|\gamma^{(t-1)})$

$\gamma^{(t)} \leftarrow \texttt{flip}(\gamma^{(t-1)}|i^{(t)})$ where $\texttt{flip}(\gamma|i)$ flips the $i$-th coordinate of $\gamma : \gamma_i \leftarrow 1 - \gamma_i$.

Sample $\mathcal{S}^{(t)} \sim \mathcal{U}(\cdot|i^{(t)}, \mathcal{A})$

Compute the unnormalized weights $\tilde{\rho}^{(t)} \leftarrow \phi^{-1}(\gamma^{(t)}, \mathcal{S}^{(t)})$

**if** $t \leq T_{\text{burn}}$ **then**

Adapt $\mathcal{A}$ using some adaptive scheme.

**end if**

**end for**

**for** $t = 1, 2, \cdots, T$ **do**

$\rho^{(t)} \leftarrow \frac{\tilde{\rho}^{(t)}}{\sum_{s > T_{\text{burn}}}^{T} \tilde{\rho}^{(s)}}$

**end for**

**Output:** $\{\rho^{(t)}, \gamma^{(t)}\}_{t=1}^{T}$.

---

The details of this algorithm is described in ALG 1. The associated estimator for this sampler is defined as (Jankowiak, 2023):

$$\texttt{PIP}(i) := \sum_{t=1}^{T} \rho^{(t)} \big( \mathbf{1}\{i \in \mathcal{S}^{(t)}\} p(\gamma_i^{(t)} = 1|\gamma_{-i}^{(t)}, \mathcal{D}) + \mathbf{1}\{i \notin \mathcal{S}^{(t)}\} \gamma_i^{(t)} \big). \tag{14}$$

## 3.2 A Variable Complexity wTGS Scheme

In the subset wTGS in Subsection 3.1, the number of conditional PIP computations per MCMC iteration is fixed, i.e., it is equal to $S$. In the following, we propose a variable complexity-based wTGS scheme (VC-wTGS), say ALG 2, where the only requirement is that the expected number of the conditional PIP computations per MCMC iteration is $S$. This means that $\mathbb{E}[S_t] = S$, where $S_t$ is the number of conditional PIP computations at the $t$-th MCMC iteration.

Compared with ALG 1, ALG 2 allows us to use different subset sizes at MCMC iterations. By ALG 2, the expectation of number of conditional PIP computations in each MCMC iteration is $P \times (S/P) + 0 \times (1 - S/P) = S$. Since we aim to bound the variance at each finite iteration $T$, we don't mention about $T_{\text{burn}}$ in ALG 2. In practice, we usually remove some initial samples. We also use the following new version of Rao-Blackwellized

estimator:

$$\text{PIP}(i) := \sum_{t=1}^{T} \rho^{(t)} p(\gamma_i^{(t)} = 1 | \gamma_{-i}^{(t)}, \mathcal{D}). \tag{15}$$

In ALG 2, Bernoulli random variables $\{Q^{(t)}\}_{t=1}^{T}$ are used to replace for random set $\mathcal{S}$ in ALG 1. There are

---

**Algorithm 2** A Variable-Complexity Based wTGS Algorithm

---

**Input:** Dataset $\mathcal{D} = \{X, Y\}$ with $P$ covariates; prior inclusion probability $h$; prior precision $\tau$; total number of MCMC iterations $T$; subset size $S$.
**Output:** Approximate weighted posterior samples $\{\rho^{(t)}, \gamma^{(t)}\}_{t=1}^{T}$
**Initializations:** $\gamma^{(0)} = (\gamma_1, \gamma_2, \cdots, \gamma_P)$ where $\gamma_j \sim \texttt{Bern}(h)$ for all $j \in [P]$.
**for** $t = 1, 2, \cdots, T$ **do**
   Set $Q^{(1)} = 1$. Sample a Bernoulli random variable $Q^{(t)} \sim \texttt{Bern}(\frac{S}{P})$ if $t \geq 2$.
   **if** $Q^{(t)} = 1$ **then**
      Estimate $P$ conditional PIPs $p(\gamma_j^{(t-1)} | \gamma_{-j}^{(t-1)}, \mathcal{D})$ for all $j \in [P]$
      $\phi(\gamma^{(t-1)}) \leftarrow \sum_{j \in [P]} \frac{\frac{1}{2}\eta(\gamma_{-j}^{(t-1)})}{p(\gamma_j^{(t-1)} | \gamma_{-j}^{(t-1)}, \mathcal{D})}$
      Estimate $f(j | \gamma^{(t-1)}) \leftarrow \phi^{-1}(\gamma^{(t-1)}) \frac{\frac{1}{2}\eta(\gamma_{-j}^{(t-1)})}{p(\gamma_j^{(t-1)} | \gamma_{-j}^{(t-1)}, \mathcal{D})}$ for all $j \in [P]$.
      Sample $i^{(t)} \sim f(\cdot | \gamma^{(t-1)})$
      $\gamma^{(t)} \leftarrow \texttt{flip}(\gamma^{(t-1)} | i^{(t)})$ where $\texttt{flip}(\gamma | i)$ flips the $i$-th coordinate of $\gamma : \gamma_i \leftarrow 1 - \gamma_i$.
      Compute the unnormalized weights $\tilde{\rho}^{(t)} \leftarrow \phi^{-1}(\gamma^{(t)})$
   **else**
      $\gamma^{(t)} \leftarrow \gamma^{(t-1)}$
      $\tilde{\rho}^{(t)} \leftarrow \phi^{-1}(\gamma^{(t)})$
   **end if**
**end for**
**for** $t = 1, 2, \cdots, T$ **do**
   $\rho^{(t)} \leftarrow \frac{\tilde{\rho}^{(t)} Q^{(t)}}{\sum_{s=1}^{T} \tilde{\rho}^{(s)} Q^{(s)}}$
**end for**
**Output:** $\{\rho^{(t)}, \gamma^{(t)}\}_{t=1}^{T}$.

---

two main reasons for this replacement: (1) generating a random set $\mathcal{S}$ from $\binom{P}{S}$ subsets of $[P]$ takes very long running time for most pairs $(P, S)$, (2) the associated Rao-Blackwellized estimator usually has smaller variance with ALG 2 than ALG 1 at high $P/S$. See Section 4 for our simulation results.

### 3.3 Theoretical Bounds for Algorithm 2

First, we prove the following result. The proof can be found in Appendix C.

**Lemma 1.** *Let $U$ and $V$ be two positive random variables such that $U/V \leq M$ a.s. for some constant $M$. In addition, assume that on a set $D$ with probability at least $1 - \alpha$, we have*

$$|U - \mathbb{E}[U]| \leq \varepsilon \mathbb{E}[U], \tag{16}$$
$$|V - \mathbb{E}[V]| \leq \varepsilon \mathbb{E}[V], \tag{17}$$

*for some $0 \leq \varepsilon < 1$. Then, it holds that*

$$\mathbb{E}\left[\left|\frac{U}{V} - \frac{\mathbb{E}[U]}{\mathbb{E}[V]}\right|^2\right] \leq \frac{4\varepsilon^2}{(1-\varepsilon)^2}\left(\frac{\mathbb{E}[U]}{\mathbb{E}[V]}\right)^2 + \left[\max\left(M, \frac{\mathbb{E}[U]}{\mathbb{E}[V]}\right)\right]^2 \alpha. \tag{18}$$

We also recall the following Hoeffding's inequality for Markov chain:

**Lemma 2.** *(Rao, 2018, Theorem 1.1) Let $\{Y_i\}_{i=1}^{\infty}$ be a stationary Markov chain with state space $[N]$, transition matrix $A$, stationary probability measure $\pi$, and averaging operator $E_\pi$, so that $Y_1$ is distributed according to $\pi$. Let $\lambda = \|A - E_\pi\|_{L_2(\pi) \to L_2(\pi)}$ and let $f_1, f_2, \cdots, f_n : [N] \to \mathbb{R}$ so that $\mathbb{E}[f_i(Y_i)] = 0$ for all $i$ and $|f_i(\nu)| \le a_i$ for all $\nu \in [N]$ and all $i$. Then for $u \ge 0$,*

$$\mathbb{P}\left[\left|\sum_{i=1}^{n} f_i(Y_i)\right| \ge u\left(\sum_{i=1}^{n} a_i^2\right)^{1/2}\right] \le 2\exp\left(-\frac{u^2(1-\lambda)}{64e}\right). \tag{19}$$

Now, the following result can be shown.

**Lemma 3.** *Let*

$$\phi(\gamma) := \sum_{j \in [P]} \frac{\frac{1}{2}\eta(\gamma_{-j})}{p(\gamma_j | \gamma_{-j}, \mathcal{D})} \tag{20}$$

*and define*

$$f(\gamma) := \phi(\gamma)p(\gamma|\mathcal{D}). \tag{21}$$

*Then, by ALG 2, the sequence $\{\gamma^{(t)}, Q^{(t)}\}_{t=1}^{T}$ forms a reversible Markov chain with the stationary distribution proportional to $f(\gamma)q(Q)$ where $q$ is the Bernoulli $(S/P)$ distribution. This Markov chain has transition kernel $K((\gamma, Q) \to (\gamma', Q')) = K^*(\gamma \to \gamma')q(Q')$ where*

$$K^*(\gamma \to \gamma') = \frac{S}{P}\sum_{j=1}^{P} f(j|\gamma)\delta(\gamma' - \textit{flip}(\gamma|j)) + \left(1 - \frac{S}{P}\right)\delta(\gamma' - \gamma). \tag{22}$$

In the classical wTGS (Zanella & Roberts, 2019), the Markov chain $\{\gamma^{(t)}\}_{t=1}^{T}$ also form a Markov chain. However, this Markov chain is different from the Markov chain in Lemma 3. However, the two Markov chains still have the same stationary distribution which is proportional to $f(\gamma)$. See a detailed proof of Lemma 3 in Appendix B.

**Lemma 4.** *For the Rao-Blackwellized estimator in* (15) *which is applied to the output sequence $\{\rho^{(t)}, \gamma^{(t)}\}_{t=1}^{T}$ of ALG 2, it holds that*

$$E_{i,T} := \sum_{t=1}^{T} \rho^{(t)} p(\gamma_i^{(t)} = 1 | \gamma_{-i}^{(t)}, \mathcal{D}) \to \textit{PIP}(i) \tag{23}$$

*as $T \to \infty$.*

*Proof.* By Lemma 3, $\{\gamma^{(t)}, Q^{(t)}\}_{t=1}^{T}$ forms a reversible Markov chain with stationary distribution $f(\gamma)/Z_f q(Q)$ where $Z_f = \sum_\gamma f(\gamma)$. Hence, by SLLN for Markov chain (Breiman, 1960), for any bounded function $h$, we have

$$\frac{1}{T}\sum_{t=1}^{T} \phi^{-1}(\gamma^{(t)})Q^{(t)}h(\gamma^{(t)})$$

$$\to \mathbb{E}_{qf(\cdot)/Z_f}\left[\phi^{-1}(\gamma)h(\gamma)Q\right] \tag{24}$$

$$= \sum_Q q(Q)\sum_\gamma \frac{f(\gamma)}{Z_f}\phi^{-1}(\gamma)h(\gamma)Q \tag{25}$$

$$= \left(\sum_Q q(Q)Q\right)\left(\sum_\gamma \frac{f(\gamma)}{Z_f}\phi^{-1}(\gamma)h(\gamma)\right) \tag{26}$$

$$= \mathbb{E}_q[Q]\frac{1}{Z_f}\sum_\gamma p(\gamma|\mathcal{D})h(\gamma) \tag{27}$$

$$= \frac{S}{P}\frac{1}{Z_f}\sum_\gamma p(\gamma|\mathcal{D})h(\gamma), \tag{28}$$

where (27) follows from $f(\gamma) = p(\gamma|\mathcal{D})\phi(\gamma)$.

Similarly, we have

$$\frac{1}{T}\sum_{t=1}^{T}Q^{(t)}\phi^{-1}(\gamma^{(t)})$$

$$\rightarrow \mathbb{E}_{qf(\cdot)/Z_f}\left[\phi^{-1}(\gamma)Q\right] \tag{29}$$

$$= \sum_{Q}q(Q)Q\sum_{\gamma}\frac{f(\gamma)}{Z_f}\phi^{-1}(\gamma) \tag{30}$$

$$= \mathbb{E}_q[Q]\sum_{\gamma}\frac{1}{Z_f}p(\gamma|\mathcal{D}) \tag{31}$$

$$= \frac{S}{P}\frac{1}{Z_f}, \tag{32}$$

where (31) also follows from $f(\gamma) = p(\gamma|D)\phi(\gamma)$. $\qquad\square$

From (28) and (32), we obtain

$$\frac{\frac{1}{T}\sum_{t=1}^{T}\phi^{-1}(\gamma^{(t)})Q^{(t)}h(\gamma^{(t)})}{\frac{1}{T}\sum_{t=1}^{T}Q^{(t)}\phi^{-1}(\gamma^{(t)})} \rightarrow \sum_{\gamma}p(\gamma|\mathcal{D})h(\gamma), \tag{33}$$

or equivalently

$$\sum_{t=1}^{T}\rho^{(t)}h(\gamma^{(t)}) \rightarrow \sum_{\gamma}p(\gamma|\mathcal{D})h(\gamma) \tag{34}$$

as $T \rightarrow \infty$.

Now, by setting $h(\gamma) = p(\gamma_i = 1|\gamma_{-i}, \mathcal{D})$, from (34), we obtain

$$\sum_{t=1}^{T}\rho^{(t)}p(\gamma_i^{(t)} = 1|\gamma_{-i}^{(t)}, \mathcal{D}) \rightarrow \mathtt{PIP}(i) \tag{35}$$

for all $i \in [P]$.

The following result bounds the variance of PIP estimator at finite $T$.

**Lemma 5.** *For any $\varepsilon \in [0, 1]$, let $\nu$ and $\pi$ be the initial and stationary distributions of the reversible Markov sequence $\{(\gamma^{(t)}, Q^{(t)})\}$. Define*

$$\hat{\phi}(\gamma) := \frac{\phi^{-1}(\gamma)}{\max_\gamma \phi^{-1}(\gamma)}, \tag{36}$$

*and*

$$\varepsilon_0 = \frac{P}{PIP(i)\mathbb{E}_\pi[\hat{\phi}(\gamma)]S}\sqrt{\frac{64e\log T}{(1 - \lambda_{\gamma,Q})T}}. \tag{37}$$

*Then, we have*

$$\mathbb{E}\left[\left|\sum_{t=1}^{T}\rho^{(t)}p(\gamma_i^{(t)} = 1|\gamma_{-i}^{(t)}, \mathcal{D}) - PIP(i)\right|^2\right]$$

$$\leq \frac{4\varepsilon_0^2}{(1 - \varepsilon_0)^2}PIP^2(i) + \frac{4P}{S}\frac{1}{\min_\gamma \pi(\gamma)T} \rightarrow 0, \tag{38}$$

*as $T \rightarrow \infty$ for fixed $P, S$ and the dataset. Here, $\pi(\gamma)$ is the marginal distribution of $\pi(\gamma, Q)$.*

*Proof.* See Appendix D. □

**Remark 6.** *As in the proof of Lemma 3, we have $\pi(\gamma) \sim f(\gamma) = \phi(\gamma)p(\gamma|\mathcal{D})$. Hence, it holds that*

$$\min_{\gamma} \pi(\gamma) = \min_{\gamma} \frac{\phi(\gamma)p(\gamma|\mathcal{D})}{\sum_{\gamma} \phi(\gamma)p(\gamma|\mathcal{D})}, \tag{39}$$

*which does not depend on $S$.*

Next, we provide a lower bound for $1 - \lambda_{\gamma,Q}$. First, we recall the following Dirichlet form on spectral gap.

**Definition 7.** *Let $f, g : \Omega \to \mathbb{R}$. The Dirichlet form associated with a reversible Markov chain $Q$ on $\Omega$ is defined by*

$$\mathcal{E}(f,g) = \langle (\mathbf{I} - \mathbf{Q})f, g \rangle_{\pi} \tag{40}$$

$$= \sum_{x \in \Omega} \pi(x)[f(x) - \mathbf{Q}f(x)]g(x) \tag{41}$$

$$= \sum_{x,y \in \Omega \times \Omega} \pi(x)Q(x,y)g(x)(f(x) - f(y)). \tag{42}$$

**Lemma 8.** *(Diaconis & Saloff-Coste, 1993) (Variational characterisation) For a reversible Markov chain $Q$ with state space $\Omega$ and stationary distribution $\pi$, it holds that*

$$1 - \lambda = \inf_{\substack{g:\Omega \to \mathbb{R}, \\ \mathbb{E}_{\pi}[g]=0, \mathbb{E}_{\pi}[g^2]=1}} \mathcal{E}(g,g), \tag{43}$$

*where $\mathcal{E}(g,g) := \langle (\mathbf{I} - \mathbf{Q})g, g \rangle_{\pi}$.*

**Lemma 9.** *The spectral gap $1 - \lambda_{\gamma,Q}$ of the reversible Markov chain $\{\gamma^{(t)}, Q^{(t)}\}$ satisfies*

$$1 - \lambda_{\gamma,Q} \geq \frac{S}{P}\big(1 - \lambda_P\big) + 1 - \frac{S}{P} \geq 1 - \frac{S}{P}, \tag{44}$$

*where $1 - \lambda_P$ is the spectral gap of the reversible Markov chain $\{\gamma^{(t)}\}$ of the wTGS algorithm (i.e. $S = P$).*

See Appendix F for a proof of this lemma.

By combining Lemma 4, Lemma 5 and Lemma 9, we come up with the following theorem.

**Theorem 10.** *For the variable-complexity subset wTGS-based estimator in (15) and given dataset $(X, Y)$, it holds that*

$$E_{i,T} := \sum_{t=1}^{T} \rho^{(t)}p(\gamma_i^{(t)} = 1|\gamma_{-i}^{(t)}, \mathcal{D}) \to PIP(i) \tag{45}$$

*as $T \to \infty$ and*

$$\mathbb{E}\left[\left|\sum_{t=1}^{T} \rho^{(t)}p(\gamma_i^{(t)}|\gamma_{-i}^{(t)}, \mathcal{D}) - PIP(i)\right|^2\right]$$

$$= O\left(\frac{\log T}{T}\left(\frac{P}{S}\right)^2\left(\frac{\max_{\gamma} \phi(\gamma)}{\min_{\gamma} \phi(\gamma)}\right)^2\right), \tag{46}$$

*where*

$$\phi(\gamma) = \frac{1}{2} \sum_{j \in [P]} \frac{p(\gamma_j = 1|\gamma_{-j}, \mathcal{D})}{p(\gamma_j|\gamma_{-j}, \mathcal{D})}. \tag{47}$$

*Proof.* First, (45) is shown in Lemma 4. Now, we show (46) by using Lemma 5 and Lemma 9.

Observe that

$$\mathbb{E}_\pi[\hat{\phi}(\gamma)] = \mathbb{E}_\pi\left[\frac{\phi^{-1}(\gamma)}{\max_\gamma \phi^{-1}(\gamma)}\right]$$
$$\geq \frac{\min_\gamma \phi(\gamma)}{\max_\gamma \phi(\gamma)}. \tag{48}$$

In addition, we have

$$\phi(\gamma) = \sum_{j\in[P]} \frac{\frac{1}{2}\eta(\gamma_{-j})}{p(\gamma_j|\gamma_{-j},\mathcal{D})} \tag{49}$$
$$= \frac{1}{2}\sum_{j\in[P]} \frac{p(\gamma_j=1|\gamma_{-j},\mathcal{D})}{p(\gamma_j|\gamma_{-j},\mathcal{D})}. \tag{50}$$

Now, note that

$$\frac{p(\gamma_j=1|\gamma_{-j},\mathcal{D})}{p(\gamma_j|\gamma_{-j},\mathcal{D})} = \begin{cases} 1, & \gamma_j=1 \\ \frac{p(\gamma_j=1|\gamma_{-j},\mathcal{D})}{p(\gamma_j=0|\gamma_{-j},\mathcal{D})}, & \gamma_j=0. \end{cases} \tag{51}$$

In Appendix E, show how to estimate the conditional PIPs, i.e., $p(\gamma_i|\mathcal{D},\gamma_{-i})$ for the linear regression model. More specially, we have

$$p(\gamma_i|\mathcal{D},\gamma_{-i}) = \frac{p(\gamma_i|\mathcal{D},\gamma_{-i})}{p(1-\gamma_i|\mathcal{D},\gamma_{-i})}\left(1 + \frac{p(\gamma_i|\mathcal{D},\gamma_{-i})}{p(1-\gamma_i|\mathcal{D},\gamma_{-i})}\right)^{-1}. \tag{52}$$

Then, we can estimate $\frac{p(\gamma_j=1|\gamma_{-j},\mathcal{D})}{p(\gamma_j=0|\gamma_{-j},\mathcal{D})}$ based on the dataset. More specifically, let $\tilde{\gamma}_1$ is given by $\gamma_{-i}$ with $\gamma_i=1$, $\tilde{\gamma}_0$ is given by $\gamma_{-i}$ with $\gamma_i=0$, then we can show that

$$\frac{p(\gamma_j=1|\gamma_{-j},\mathcal{D})}{p(\gamma_j=0|\gamma_{-j},\mathcal{D})}$$
$$= \left(\frac{h}{1-h}\right)\sqrt{\tau\frac{\det(X_{\tilde{\gamma}_0}^T X_{\tilde{\gamma}_0} + \tau I)}{\det(X_{\tilde{\gamma}_1}^T X_{\tilde{\gamma}_1} + \tau I)}}$$
$$\times \left(\frac{\|Y\|^2 - \|\tilde{Y}_{\tilde{\gamma}_0}\|^2 + \nu_0\lambda_0}{\|Y\|^2 - \|\tilde{Y}_{\tilde{\gamma}_1}\|^2 + \nu_0\lambda_0}\right)^{\frac{N+\nu_0}{2}}. \tag{53}$$

Here, $\|\tilde{Y}_\gamma\|^2 = \tilde{Y}_\gamma^T\tilde{Y}_\gamma = Y^T X_\gamma(X_\gamma^T X_\gamma + \tau I)^{-1}X_\gamma^T Y$.

Using this algorithm, if pre-computing $X^T X$ is not possible, the computational complexity per conditional PIP is $O(N|\gamma|^2 + |\gamma|^3 + P|\gamma|^2)$. Otherwise, if pre-computing $X^T X$ is possible, the computational complexity per conditional PIP is $O(|\gamma|^3 + P|\gamma|^2)$. □

**Remark 11.** *As we can see in Appendix E, for the linear regression model in Section 2.2, if pre-computing $X^T X$ is not possible, the computational complexity for a conditional PIP is $O(N|\gamma|^2 + |\gamma|^3 + P|\gamma|^2)$. Otherwise, if pre-computing $X^T X$ is possible, the computational complexity for a conditional PIP is $O(|\gamma|^3 + P|\gamma|^2)$. Here, $|\gamma| \approx hP$. Hence, the average computational complexity for our algorithm is $O(S(N|\gamma|^2 + |\gamma|^3 + P|\gamma|^2))$ or $O(S(|\gamma|^3 + P|\gamma|^2))$ which depends on whether the precomputing of $X^T X$ is possible or not. To reduce the computational complexity, we can reduce $S$, or we are only interested in the case $P/S$ is large. This computational complexity reductions is more meaningful if $|\gamma| \approx Ph \ll P$, i.e., we consider the sparse linear regression regimes. However, the variance of the associated Rao-Blackwellized estimator is increased as $S$ becomes small. Hence, there is a trade-off between the computational complexity per MCMC iteration*

*vs. the variance of of the Rao-Blackwellized estimator. The most interesting fact is that the newly-designed Rao-Blackwellized estimator converges to PIPs for any value of $S$. In practice, the choice of $S$ depends on each application and the availability of computational resources. We can choose $S$ very small (eg., $S = 2$) to have a low complexity estimator and low convergence rate. We can choose $S \approx P$ for a high complexity estimator with high convergence rate. Furthermore, both our and Jankowiak algorithms are degenerated to the wTGS (Zanella & Roberts (2019)) at $S \approx P$.*

## 4    Experiments

In this section, we show by simulation that the PIP-estimator is convergent as $T \to \infty$. In addition, we compare the variance of associated Rao-Blackwellized estimators for VC-wTGS and subset wTGS on simulated and real datasets. To compute $p(\gamma_i|\gamma_{-i}, Y)$, we use the same trick as (Zanella & Roberts, 2019, Appendix B.1) for the new setting. See our derivations of this posterior distribution in Appendix E. As (Jankowiak, 2023), in ALG 1 and ALG 2, we choose

$$\eta(\gamma_{-i}) = \mathbb{P}(\gamma_i = 1|\gamma_{-i}, \mathcal{D}). \tag{54}$$

### 4.1    Simulated Datasets

First, we perform a simulated experiment. Let $X \in \mathbb{R}^{N \times P}$ be a realization of a multivariate (random) Gaussian matrix. We consider the case $N = 100$ and $P = 200$. We run $T = 20000$ iterations.

Fig. 1 shows the number of conditional PIP computations per MCMC iteration over $T$ iterations. As we can see, our algorithm (Algorithm 2) has variable complexity where the number of conditional PIP computations per MCMC is a random variable $Y$ which takes value on $\{0, P\}$ where $\mathbb{P}(Y = P) = S/P$. For Jankowiak's algorithm, the number of conditional PIP computations per MCMC is always fixed, which is equal to $S$.

Fig. 2 shows that the Rao-Blackwellized estimator in (15) converges to the value of PIP at $T \to \infty$ for different values of $S$. Since the number of PIPs, $P$, is very large, we only run simulations for PIP(0) and PIP(1). The behavior of PIP(0) and PIP(1) represents the behavior of other PIPs. Since VC-wTGS converges very fast at $T$ big enough, the variance of variable-complexity wTGS is very small in the long term.

In Fig. 4, we plot the estimators of VC-wTGS, subset wTGS, and wTGS for estimating PIP(0). It can our estimator converges to wTGS estimator faster than subset wTGS. This also means that the variance of VC-wTGS is smaller than the variance of subset wTGS for the same sample complexity $S$.

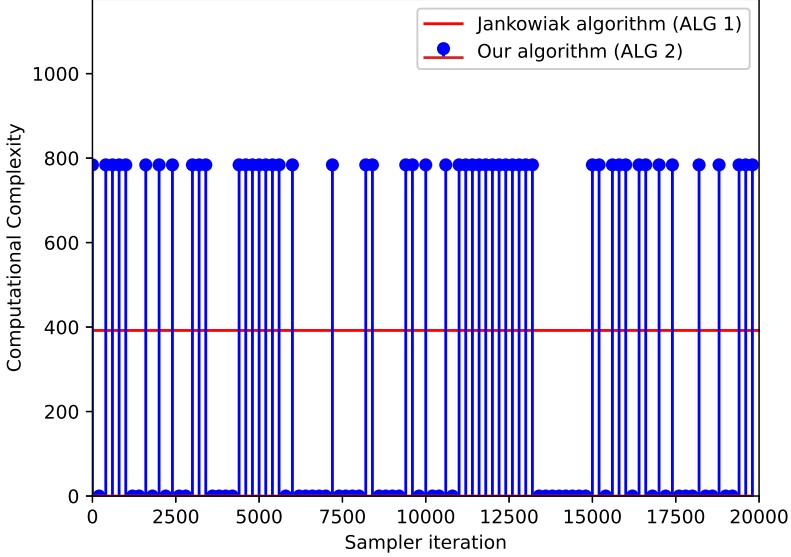

Figure 1: Computational Complexity Evolution

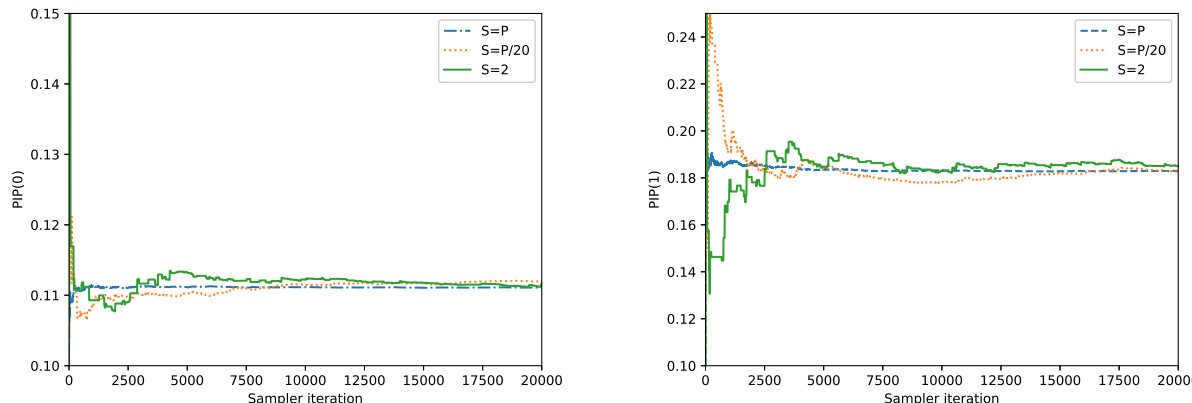

Figure 2: VC-wTGS Rao-Blackwellized Estimators (ALG 2)

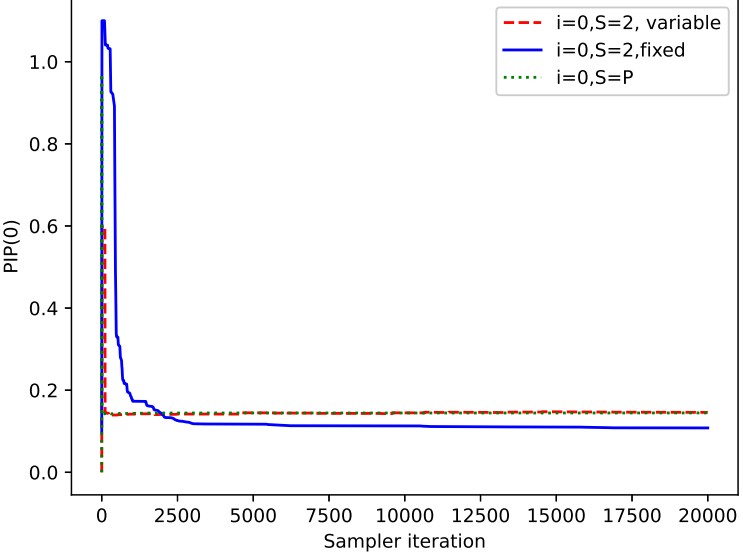

Figure 3: Convergence of Rao-Blackwellized Estimators

## 4.2 Real Datasets

In this simulation, we run ALG 2 on MNIST dataset.

As Fig. 1, Fig. 4 shows the number of conditional PIP computations per MCMC iteration over $T$ iterations. It shows that our algorithm has variable computational complexity per MCMC iteration, which is different from Jankowiak's algorithm.

Fig. 5 plots PIP(0) and PIP(1) and the estimated variances for the Rao-Blackwellized estimator in (15) at different values of $S$, respectively. Here, PIP(0) and PIP(1) are defined in (9), which are posterior inclusion probabilities that the components $\beta_0$ and $\beta_1$ affect the output. These plots show a trade-off between the computational complexity and the estimated variance for estimating PIP(0) and PIP(1). The

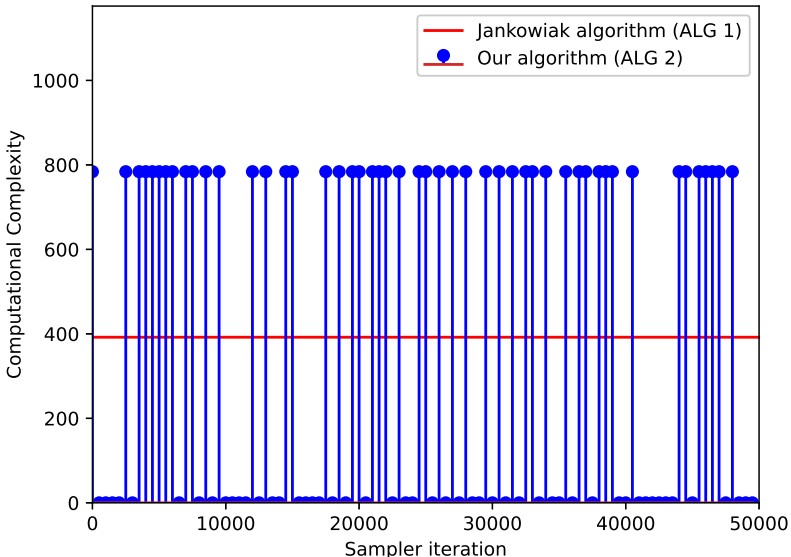

Figure 4: Computational Complexity Evolution

expected number of PIP computations is only $ST$ in ALG 2 but $TP$ in wTGS if we run $T$ MCMC iterations. However, we suffer an increasing in variance. By Theorem 10, the variance is $O\big(\big(\frac{P}{S}\big)^2 \frac{\log T}{T}\big)$ for a given dataset, i.e., increasing at most $(P/S)^2$ times. For many applications, we don't need to estimate PIPs exactly, hence VC-wTGS can be used to reduce computational complexity especially when $P$ is very large (million covariates). Fig. 6 shows that VC-wTGS outperforms subset wTGS (Jankowiak, 2023) at high values of $P/S$, which shows that our newly-designed Rao-Blackwellized estimator converges to PIP faster than Jankowiak's estimator at high $P/S$.

## 5 Conclusion

This paper proposed a variable complexity wTGS for Bayesian Variable Selection which can improve the computational complexity of the well-known wTGS. Experiments show that our Rao-Blackwellized estimator can give a smaller variance than its counterpart associated with the subset-wTGS at high $P/S$.

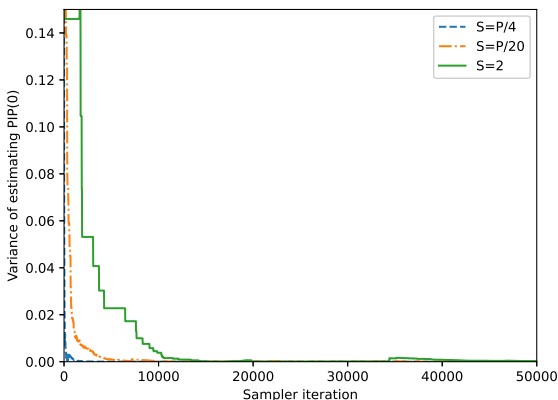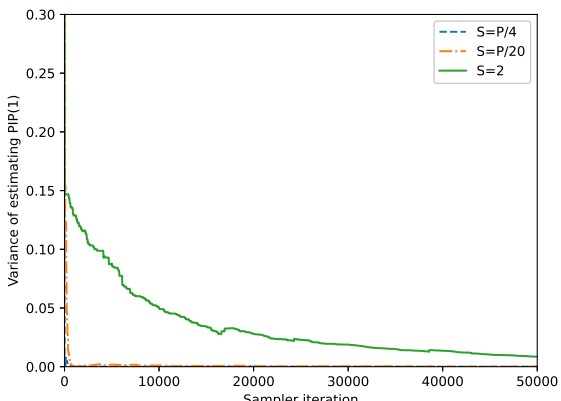

Figure 5: The variance of VC-wTGS Rao-Blackwellized Estimators (ALG 2)

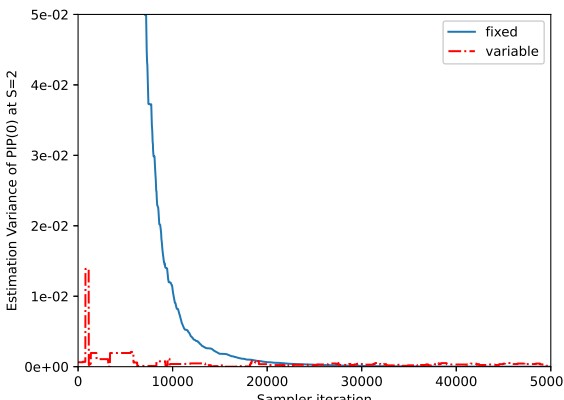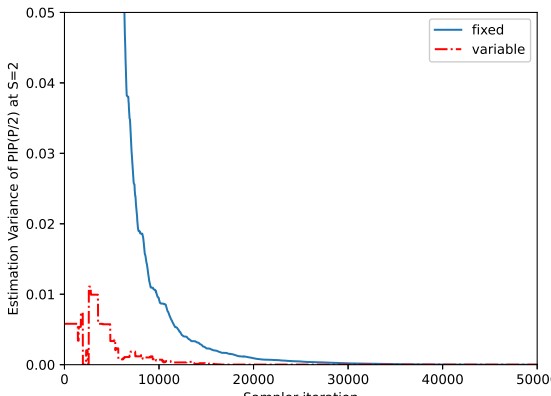

Figure 6: Comparing the variance between subset wTGS and VC-wTGS at $S = 2$.

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

## A Appendix

## B Proof of Lemma 3

The transition kernel for the sequence $\{\gamma^{(t)}\}$ can be written as

$$K^*(\gamma \to \gamma') = \frac{S}{P} \sum_{j=1}^{P} f(j|\gamma)\delta(\gamma' - \texttt{flip}(\gamma|j)) + \left(1 - \frac{S}{P}\right)\delta(\gamma' - \gamma). \tag{55}$$

This implies that for any pair $(\gamma, \gamma')$ such that $\gamma' = \texttt{flip}(\gamma|i)$ for some $i \in [P]$, we have

$$K^*(\gamma \to \gamma') = \frac{S}{P} \sum_{j=1}^{P} f(j|\gamma)\delta(\gamma' - \texttt{flip}(\gamma|j)) \tag{56}$$

$$= \frac{S}{P} f(i|\gamma). \tag{57}$$

Now, by ALG 2, we also have

$$f(i|\gamma) = \phi^{-1}(\gamma)\frac{\frac{1}{2}\eta(\gamma_{-i})}{p(\gamma_i|\gamma_{-i},\mathcal{D})} \tag{58}$$

and

$$f(i|\gamma') = \phi^{-1}(\gamma')\frac{\frac{1}{2}\eta(\gamma'_{-i})}{p(\gamma'_i|\gamma'_{-i},\mathcal{D})}. \tag{59}$$

From (58) and (59) and $\gamma_{-i} = \gamma'_{-i}$, we obtain

$$\frac{K^*(\gamma \to \gamma')}{K^*(\gamma' \to \gamma)} = \frac{\frac{S}{P}f(i|\gamma)}{\frac{S}{P}f(i|\gamma')} \tag{60}$$

$$= \frac{f(i|\gamma)}{f(i|\gamma')} \tag{61}$$

$$= \frac{\phi(\gamma')p(\gamma'|\mathcal{D})}{\phi(\gamma)p(\gamma|\mathcal{D})} \tag{62}$$

$$= \frac{f(\gamma')}{f(\gamma)}. \tag{63}$$

In addition, we also have $K^*(\gamma \to \gamma') = K^*(\gamma' \to \gamma) = 0$ if $\gamma' \neq \gamma$ and $\gamma' \neq \mathtt{flip}(\gamma|i)$ for any $i \in [P]$. Furthermore, $K^*(\gamma \to \gamma') = K^*(\gamma' \to \gamma) = 1 - \frac{S}{P}$ if $\gamma = \gamma'$.

By combining all these cases, it holds that

$$f(\gamma)K^*(\gamma \to \gamma') = f(\gamma')K^*(\gamma' \to \gamma) \tag{64}$$

for all $\gamma', \gamma$.

This means that $\{\gamma^{(t)}\}_{t=1}^T$ form a reversible Markov chain with stationary distribution $f(\gamma)/Z_f$ where

$$Z_f = \sum_\gamma f(\gamma). \tag{65}$$

Since $\{Q_t\}_{t=1}^T$ is an i.i.d. Bernoulli sequence with $q(1) = S/P$ and independent of $\{\gamma^{(t)}\}_{t=1}^T$, $\{\gamma^{(t)}, Q^{(t)}\}_{t=1}^T$ forms a Markov chain with the transition kernel satisfying:

$$K((\gamma, Q) \to (\gamma', Q')) = q(Q')K^*(\gamma \to \gamma'). \tag{66}$$

It follows from (66) that

$$q(Q)f(\gamma)/Z_f K((\gamma, Q) \to (\gamma', Q')) = \left[K^*(\gamma \to \gamma')f(\gamma)/Z_f\right]q(Q)q(Q') \tag{67}$$

for any pair $(\gamma, Q)$ and $(\gamma', Q')$.

Finally, from (64) and (67), we have

$$q(Q)f(\gamma)/Z_f K((\gamma, Q) \to (\gamma', Q')) = q(Q')f(\gamma)/Z_f K((\gamma', Q') \to (\gamma, Q)). \tag{68}$$

This means that $\{\gamma_t, Q^{(t)}\}_{t=1}^T$ forms a reversible Markov chain with stationary distribution $q(Q)f(\gamma)/Z_f$.

## C   Proof of Lemma 1

Observe that with probability at least $1 - \alpha$, we have

$$(1 - \varepsilon)\mathbb{E}[U] \leq U \leq (1 + \varepsilon)\mathbb{E}[U] \tag{69}$$

$$(1 - \varepsilon)\mathbb{E}[V] \leq V \leq (1 + \varepsilon)\mathbb{E}[V]. \tag{70}$$

Hence, we have

$$\left(\frac{1-\varepsilon}{1+\varepsilon}\right)\frac{\mathbb{E}[U]}{\mathbb{E}[V]} \leq \frac{U}{V} \leq \left(\frac{1+\varepsilon}{1-\varepsilon}\right)\frac{\mathbb{E}[U]}{\mathbb{E}[V]}. \tag{71}$$

From (71), with probability at least $1 - \alpha$, we have

$$\left|\frac{U}{V} - \frac{\mathbb{E}[U]}{\mathbb{E}[V]}\right| \leq \frac{2\varepsilon}{1-\varepsilon}\left(\frac{\mathbb{E}[U]}{\mathbb{E}[V]}\right). \tag{72}$$

It follows from (72) that

$$\mathbb{E}\left[\left|\frac{U}{V} - \frac{\mathbb{E}[U]}{\mathbb{E}[V]}\right|^2\right] = \mathbb{E}\left[\left|\frac{U}{V} - \frac{\mathbb{E}[U]}{\mathbb{E}[V]}\right|^2\Big|D\right]\mathbb{P}(D) + \mathbb{E}\left[\left|\frac{U}{V} - \frac{\mathbb{E}[U]}{\mathbb{E}[V]}\right|^2\Big|D^c\right]\mathbb{P}(D^c) \tag{73}$$

$$\leq \frac{4\varepsilon^2}{(1-\varepsilon)^2}\left(\frac{\mathbb{E}[U]}{\mathbb{E}[V]}\right)^2 + \left[\max\left(M, \frac{\mathbb{E}[U]}{\mathbb{E}[V]}\right)\right]^2\alpha. \tag{74}$$

## D  Proof of Lemma 5

First, by definition of $\hat{\phi}(\gamma)$ in (36) we have

$$\rho^{(t)} = \frac{\hat{\phi}(\gamma^{(t)})Q^{(t)}}{\sum_{t=1}^T \hat{\phi}(\gamma^{(t)})Q^{(t)}}. \tag{75}$$

In addition, observe that

$$0 \leq \hat{\phi}(\gamma) \leq 1. \tag{76}$$

Now, let $g : \{0,1\}^P \to \mathbb{R}_+$ such that $g(\gamma) \leq 1$ for all $\gamma$. Then, by applying Lemma 2 and a change of measure, with probability $1 - 2\frac{d\nu}{d\pi}\exp(-\frac{\zeta^2 T(1-\lambda)}{64e})$, we have

$$\frac{1}{T}\left|\sum_{t=1}^T \hat{\phi}(\gamma^{(t)})g(\gamma^{(t)})Q^{(t)} - \mathbb{E}_\pi\left[\sum_{t=1}^T \hat{\phi}(\gamma^{(t)})g(\gamma^{(t)})Q^{(t)}\right]\right| \leq \zeta \tag{77}$$

for any $\zeta > 0$.

Similarly, by using Lemma 2, with probability at least $1 - 2\frac{d\nu}{d\pi}\exp(-\frac{\zeta^2 T(1-\lambda)}{64e})$, it holds that

$$\frac{1}{T}\left|\sum_{t=1}^T \hat{\phi}(\gamma^{(t)})Q^{(t)} - \mathbb{E}_\pi\left[\sum_{t=1}^T \hat{\phi}(\gamma^{(t)})Q^{(t)}\right]\right| \leq \zeta. \tag{78}$$

By using the union bound, with probability at least $1 - 4\frac{d\nu}{d\pi}\exp(-\frac{\zeta^2 T(1-\lambda)}{64e})$, it holds that

$$\frac{1}{T}\left|\sum_{t=1}^T \hat{\phi}(\gamma^{(t)})g(\gamma^{(t)})Q^{(t)} - \mathbb{E}_\pi\left[\sum_{t=1}^T \hat{\phi}(\gamma^{(t)})g(\gamma^{(t)})Q^{(t)}\right]\right| \leq \zeta, \tag{79}$$

$$\frac{1}{T}\left|\sum_{t=1}^T \hat{\phi}(\gamma^{(t)}) - \mathbb{E}_\pi\left[\sum_{t=1}^T \hat{\phi}(\gamma^{(t)})\right]\right| \leq \zeta. \tag{80}$$

Now, by setting $\zeta = \zeta_0 := \frac{\varepsilon}{T} \min \left\{ \mathbb{E}_\pi \left[ \sum_{t=1}^T \hat{\phi}(\gamma^{(t)}) g(\gamma^{(t)}) Q^{(t)} \right], \mathbb{E}_\pi \left[ \sum_{t=1}^T \hat{\phi}(\gamma^{(t)}) \right] \right\}$ for some $\varepsilon > 0$ (to be chosen later), with probability at least $1 - 4\frac{d\nu}{d\pi} \exp(-\frac{\zeta_0^2 T(1-\lambda)}{64e})$, it holds that

$$\frac{1}{T} \left| \sum_{t=1}^T \hat{\phi}(\gamma^{(t)}) g(\gamma^{(t)}) Q^{(t)} - \mathbb{E}_\pi \left[ \sum_{t=1}^T \hat{\phi}(\gamma^{(t)}) g(\gamma^{(t)}) Q^{(t)} \right] \right|$$

$$\leq \frac{\varepsilon}{T} \mathbb{E}_\pi \left[ \sum_{t=1}^T \hat{\phi}(\gamma^{(t)}) g(\gamma^{(t)}) Q^{(t)} \right], \tag{81}$$

$$\frac{1}{T} \left| \sum_{t=1}^T \hat{\phi}(\gamma^{(t)}) Q^{(t)} - \mathbb{E}_\pi \left[ \sum_{t=1}^T \hat{\phi}(\gamma^{(t)}) Q^{(t)} \right] \right| \leq \frac{\varepsilon}{T} \mathbb{E}_\pi \left[ \sum_{t=1}^T \hat{\phi}(\gamma^{(t)}) Q^{(t)} \right]. \tag{82}$$

Furthermore, by setting

$$U := \frac{1}{T} \sum_{t=1}^T \hat{\phi}(\gamma^{(t)}) g(\gamma^{(t)}) Q^{(t)}, \tag{83}$$

$$V := \frac{1}{T} \sum_{t=1}^T \hat{\phi}(\gamma^{(t)}) Q^{(t)}, \tag{84}$$

we have

$$\frac{U}{V} = \frac{\sum_{t=1}^T \phi^{-1}(\gamma^{(t)}) g(\gamma^{(t)}) Q^{(t)}}{\sum_{t=1}^T \phi^{-1}(\gamma^{(t)}) Q^{(t)}} \tag{85}$$

$$= \sum_{t=1}^T \rho^{(t)} g(\gamma^{(t)}) \tag{86}$$

and

$$M := \sup(U/V) \leq 1 \tag{87}$$

since $\sum_{t=1}^T \rho^{(t)} = 1$ and $g(\gamma^{(t)}) \leq 1$ for all $\gamma^{(t)}$.

From (80)-(87), by Lemma 1, we have

$$\mathbb{E} \left[ \left| \sum_{t=1}^T \rho^{(t)} g(\gamma^{(t)}) Q^{(t)} - \frac{\mathbb{E}_\pi[U]}{\mathbb{E}_\pi[V]} \right|^2 \right] \leq \frac{4\varepsilon^2}{(1-\varepsilon)^2} \left( \frac{\mathbb{E}_\pi[U]}{\mathbb{E}_\pi[V]} \right)^2 + \left[ \max \left( 1, \frac{\mathbb{E}_\pi[U]}{\mathbb{E}_\pi[V]} \right) \right]^2 \alpha, \tag{88}$$

where $\alpha := 4\frac{d\nu}{d\pi} \exp \left( -\frac{\varepsilon^2 T(1-\lambda_{\gamma,Q}) \min\{\mathbb{E}_\pi[U], \mathbb{E}_\pi[V]\}^2}{64e} \right)$, where $\lambda_{\gamma,Q}$ is the stationary distribution of the reversible Markov chain $\{\gamma^{(t)}, Q^{(t)}\}$.

Now, by setting

$$\varepsilon = \varepsilon_0 = \frac{1}{\min\{\mathbb{E}_\pi[U], \mathbb{E}_\pi[V]\}} \sqrt{\frac{64e \log T}{(1-\lambda_{\gamma,Q})T}}, \tag{89}$$

we have $\alpha = 4\frac{d\nu}{d\pi} \frac{1}{T}$. Then, we obtain

$$\mathbb{E} \left[ \left| \sum_{t=1}^T \rho^{(t)} g(\gamma^{(t)}) - \frac{\mathbb{E}_\pi[U]}{\mathbb{E}_\pi[V]} \right|^2 \right] \leq \frac{4\varepsilon_0^2}{(1-\varepsilon_0)^2} \left( \frac{\mathbb{E}_\pi[U]}{\mathbb{E}_\pi[V]} \right)^2 + \left[ \max \left( 1, \frac{\mathbb{E}_\pi[U]}{\mathbb{E}_\pi[V]} \right) \right]^2 \alpha. \tag{90}$$

Now, observe that

$$\frac{\mathbb{E}_\pi[U]}{\mathbb{E}_\pi[V]} = \frac{\mathbb{E}_\pi \left[ g(\gamma) Q \hat{\phi}(\gamma) \right]}{\mathbb{E}_\pi \left[ \hat{\phi}(\gamma) Q \right]} \tag{91}$$

$$= \frac{\mathbb{E}_\pi \left[ g(\gamma) Q \phi^{-1}(\gamma) \right]}{\mathbb{E}_\pi \left[ \phi^{-1}(\gamma) Q \right]}. \tag{92}$$

On the other hand, by Lemma 3, we have $\pi(\gamma, Q) = \frac{q(Q)f(\gamma)}{Z_f}$ where $Z_f := \sum_\gamma f(\gamma)$ and $f(\gamma) = p(\gamma|\mathcal{D})\phi(\gamma)$. It follows that

$$\mathbb{E}_\pi \left[g(\gamma)Q\phi^{-1}(\gamma)\right] = \mathbb{E}_{q(Q)f(\gamma)/Z_f} \left[g(\gamma)Q\phi^{-1}(\gamma)\right] \tag{93}$$

$$= \sum_\gamma \sum_Q g(\gamma)Q\phi^{-1}(\gamma)\frac{f(\gamma)}{Z_f}q(Q) \tag{94}$$

$$= \frac{1}{Z_f} \sum_\gamma \sum_Q g(\gamma)q(Q)Qp(\gamma|\mathcal{D}) \tag{95}$$

$$= \frac{1}{Z_f} \mathbb{E}_{p(\gamma|\mathcal{D})} \left[g(\gamma)\right] \mathbb{E}_q[Q]. \tag{96}$$

Similarly, we have

$$\mathbb{E}_\pi \left[\phi^{-1}(\gamma)Q\right] = \mathbb{E}_{q(Q)f(\gamma)/Z_f} \left[\phi^{-1}(\gamma)Q\right] \tag{97}$$

$$= \sum_Q \sum_\gamma \phi^{-1}(\gamma)Q\frac{f(\gamma)}{Z_f}q(Q) \tag{98}$$

$$= \frac{1}{Z_f} \left(\sum_\gamma P(\gamma|\mathcal{D})\right) \mathbb{E}_q[Q]. \tag{99}$$

From (92), (96) and (99), we obtain

$$\frac{\mathbb{E}_\pi[U]}{\mathbb{E}_\pi[V]} = \mathbb{E}_{p(\gamma|\mathcal{D})} \left[g(\gamma)\right]. \tag{100}$$

For the given problem, by setting $g(\gamma) = p(\gamma_i = 1|\gamma_{-i}, \mathcal{D})$, from (100), we have

$$\frac{\mathbb{E}_\pi[U]}{\mathbb{E}_\pi[V]} = \texttt{PIP}(i). \tag{101}$$

In addition, we have

$$\mathbb{E}_\pi[V] = \mathbb{E}_\pi \left[\hat{\phi}(\gamma)Q\right] \tag{102}$$

$$= \sum_{\gamma,Q} \hat{\phi}(\gamma)Q\frac{f(\gamma)}{Z_f}q(Q) \tag{103}$$

$$= \left(\sum_\gamma \hat{\phi}(\gamma)\frac{f(\gamma)}{Z_f}\right)\left(\sum_Q Qq(Q)\right) \tag{104}$$

$$= \mathbb{E}_\pi[\hat{\phi}(\gamma)]\mathbb{E}_Q[Q] \tag{105}$$

$$= \frac{S}{P}\mathbb{E}_\pi[\hat{\phi}(\gamma)]. \tag{106}$$

Hence, we obtain

$$\min\{\mathbb{E}_\pi[U], \mathbb{E}_\pi[V]\} = \mathbb{E}_\pi[V] \min\left\{1, \frac{\mathbb{E}_\pi[U]}{\mathbb{E}_\pi[V]}\right\} \tag{107}$$

$$= \mathbb{E}_\pi[V] \min\left\{1, \texttt{PIP}(i)\right\} \tag{108}$$

$$= \mathbb{E}_\pi[V]\texttt{PIP}(i) \tag{109}$$

$$= \frac{S}{P}\mathbb{E}_\pi[\hat{\phi}(\gamma)]\texttt{PIP}(i). \tag{110}$$

From (90), (101), and (110), we have

$$\mathbb{E}\left[\left|\sum_{t=1}^{T}\rho^{(t)}p(\gamma_i^{(t)}=1|\gamma_{-i}^{(t)},\mathcal{D})-\mathtt{PIP}(i)\right|^2\right] \leq \frac{4\varepsilon_0^2}{(1-\varepsilon_0)^2}\mathtt{PIP}^2(i)+4\frac{d\nu}{d\pi}\frac{1}{T},\tag{111}$$

and

$$\varepsilon_0 = \frac{P}{\mathtt{PIP}(i)\mathbb{E}_\pi[\hat{\phi}(\gamma)]S}\sqrt{\frac{64e\log T}{(1-\lambda_{\gamma,Q})T}}.\tag{112}$$

Now, observe that

$$\frac{d\nu}{d\pi}(\gamma,Q) = \frac{p_{\gamma_1,Q_1}(\gamma,Q)}{\pi(\gamma,Q)}\tag{113}$$

$$\leq \frac{1}{\pi(\gamma,Q)}\tag{114}$$

$$= \frac{1}{\pi(\gamma)q(Q)}\tag{115}$$

$$\leq \frac{P}{S}\frac{1}{\min_\gamma\pi(\gamma)}.\tag{116}$$

By combining (111) and (116), we have

$$\mathbb{E}\left[\left|\sum_{t=1}^{T}\rho^{(t)}p(\gamma_i^{(t)}=1|\gamma_{-i}^{(t)},\mathcal{D})-\mathtt{PIP}(i)\right|^2\right] \leq \frac{4\varepsilon_0^2}{(1-\varepsilon_0)^2}\mathtt{PIP}^2(i)+\frac{4P}{S}\frac{1}{\min_\gamma\pi(\gamma)T}.\tag{117}$$

## E Derive $p(\gamma_i|\mathcal{D},\gamma_{-i})$

Observe that

$$p(\gamma_i|\mathcal{D},\gamma_{-i}) = \frac{p(\gamma_i|\mathcal{D},\gamma_{-i})}{p(1-\gamma_i|\mathcal{D},\gamma_{-i})}\left(1+\frac{p(\gamma_i|\mathcal{D},\gamma_{-i})}{p(1-\gamma_i|\mathcal{D},\gamma_{-i})}\right)^{-1}.\tag{118}$$

In addition, we have

$$\frac{p(\gamma_i=1|\mathcal{D},\gamma_{-i})}{p(\gamma_i=0|\mathcal{D},\gamma_{-i})} = \frac{p(\gamma_i=1,\mathcal{D}|\gamma_{-i})}{p(\gamma_i=0,\mathcal{D}|\gamma_{-i})}\tag{119}$$

$$= \frac{p(\gamma_i=1|\gamma_{-i},X)}{p(\gamma_i=0|\gamma_{-i},X)}\frac{p(Y|\gamma_i=1,\gamma_{-i},X)}{p(Y|\gamma_i=0,\gamma_{-i},X)}\tag{120}$$

$$= \left(\frac{p(\gamma_i=1)}{p(\gamma_i=0)}\right)\left(\frac{p(Y|\gamma_i=1,\gamma_{-i},X)}{p(Y|\gamma_i=0,\gamma_{-i},X)}\right)\tag{121}$$

$$= \left(\frac{h}{1-h}\right)\left(\frac{p(Y|\gamma_i=1,\gamma_{-i},X)}{p(Y|\gamma_i=0,\gamma_{-i},X)}\right).\tag{122}$$

On the other hand, for any tuple $\gamma = (\gamma_1,\gamma_2,\cdots,\gamma_P)$ such that $\gamma_i=1$ (so $|\gamma|\geq 1$), we have

$$p(Y|\gamma_i=1,\gamma_{-i},\beta_\gamma,\sigma_\gamma^2,X) = \frac{1}{\left(\sigma_\gamma\sqrt{2\pi}\right)^N}\exp\left(-\frac{\|Y-X_\gamma\beta_\gamma\|^2}{2\sigma_\gamma^2}\right).\tag{123}$$

It follows that

$$
p(Y|\gamma_i = 1, \gamma_{-i}, X)
$$

$$
= \int_{\beta_\gamma} \int_{\sigma_\gamma^2=0}^{\infty} \frac{1}{(\sigma_\gamma\sqrt{2\pi})^N} \exp\left(-\frac{\|Y - X_\gamma\beta_\gamma\|^2}{2\sigma_\gamma^2}\right) p(\beta_\gamma|\gamma_i = 1, \gamma_{-i}) p(\sigma_\gamma^2|\gamma_i = 1, \gamma_{-i}) d\beta_\gamma d\sigma_\gamma^2 \tag{124}
$$

$$
= \int_{\sigma_\gamma^2=0}^{\infty} \texttt{InvGamma}\left(\frac{1}{2}\nu_0, \frac{1}{2}\nu_0\lambda_0\right) \int_{\beta_\gamma} \frac{1}{(\sigma_\gamma\sqrt{2\pi})^N} \exp\left(-\frac{\|Y - X_\gamma\beta_\gamma\|^2}{2\sigma_\gamma^2}\right)
$$

$$
\times \frac{1}{(\sigma_\gamma\sqrt{2\pi\tau^{-1}})^{|\gamma|}} \exp\left(-\frac{\|\beta_\gamma\|^2}{2\sigma_\gamma^2\tau^{-1}}\right) d\beta_\gamma d\sigma_\gamma^2. \tag{125}
$$

Now, observe that

$$
\|Y - X_\gamma\beta_\gamma\|^2 + \tau\|\beta_\gamma\|^2
$$

$$
= (Y - X_\gamma\beta_\gamma)^T(Y - X_\gamma\beta_\gamma) + \tau\beta_\gamma^T\beta_\gamma \tag{126}
$$

$$
= Y^TY - 2Y^TX_\gamma\beta_\gamma + \beta_\gamma^T X_\gamma^T X_\gamma\beta_\gamma + \tau\beta_\gamma^T\beta_\gamma \tag{127}
$$

$$
= Y^TY - 2Y^TX_\gamma\beta_\gamma + \beta_\gamma^T(X_\gamma^T X_\gamma + \tau I)\beta_\gamma. \tag{128}
$$

Now, consider the EVD (singular value decomposition) of the positive definite matrix $X_\gamma^T X_\gamma + \tau I$ (note that $\tau > 0$):

$$
X_\gamma^T X_\gamma + \tau I = U^T\Lambda U \tag{129}
$$

where $\Lambda$ is the a diagonal matrix consisting of all positive eigenvalue of $X_\gamma^T X_\gamma + \tau I$. Let

$$
\tilde{\beta}_\gamma := \sqrt{\Lambda}U\beta_\gamma, \tag{130}
$$

$$
\tilde{Y}_\gamma := \sqrt{\Lambda^{-1}}UX_\gamma^TY. \tag{131}
$$

Then, we have

$$
\|Y - X_\gamma\beta_\gamma\|^2 + \tau\|\beta_\gamma\|^2
$$

$$
= Y^TY - 2Y^TX_\gamma\beta_\gamma + \beta_\gamma^T(X_\gamma^T X_\gamma + \tau I)\beta_\gamma \tag{132}
$$

$$
= Y^TY - 2Y^TX_\gamma\sqrt{\Lambda^{-1}}U^T\tilde{\beta}_\gamma + \tilde{\beta}_\gamma^T\tilde{\beta}_\gamma \tag{133}
$$

$$
= Y^TY - 2\tilde{Y}_\gamma^T\tilde{\beta}_\gamma + \tilde{\beta}_\gamma^T\tilde{\beta}_\gamma \tag{134}
$$

$$
= \left(\|Y\|^2 - |\tilde{Y}_\gamma|^2\right) + \left(\tilde{Y}_\gamma^T\tilde{Y}_\gamma - 2\tilde{Y}_\gamma^T\tilde{\beta}_\gamma + \tilde{\beta}_\gamma^T\tilde{\beta}_\gamma\right) \tag{135}
$$

$$
= \left(\|Y\|^2 - |\tilde{Y}_\gamma|^2\right) + \|\tilde{Y}_\gamma - \tilde{\beta}_\gamma\|^2. \tag{136}
$$

Hence, we have

$$
d\beta_\gamma = \det(U^T\Lambda^{-1/2})d\tilde{\beta}_\gamma \tag{137}
$$

$$
= \det(X_\gamma^T X_\gamma + \tau I)^{-1/2}d\tilde{\beta}_\gamma. \tag{138}
$$

Hence, we have

$$
\int_{\beta_\gamma} \frac{1}{(\sigma_\gamma\sqrt{2\pi})^N} \exp\left(-\frac{\|Y - X_\gamma\beta_\gamma\|^2}{2\sigma_\gamma^2}\right) \frac{1}{(\sigma_\gamma\sqrt{2\pi\tau^{-1}})^{|\gamma|}} \exp\left(-\frac{\|\beta_\gamma\|^2}{2\sigma_\gamma^2\tau^{-1}}\right) d\beta_\gamma \tag{139}
$$

$$
= \int_{\tilde{\beta}_\gamma} \frac{1}{(\sigma_\gamma\sqrt{2\pi})^N} \exp\left(-\frac{\left(\|Y\|^2 - |\tilde{Y}_\gamma|^2\right) + \|\tilde{Y}_\gamma - \tilde{\beta}_\gamma\|^2}{2\sigma_\gamma^2}\right)
$$

$$
\times \frac{1}{(\sigma_\gamma\sqrt{2\pi\tau^{-1}})^{|\gamma|}} \det(X_\gamma^T X_\gamma + \tau I)^{-1/2}d\tilde{\beta}_\gamma \tag{140}
$$

$$
= \frac{1}{(\sigma_\gamma\sqrt{2\pi})^N} \tau^{|\gamma|/2} \exp\left(-\frac{\left(\|Y\|^2 - |\tilde{Y}_\gamma|^2\right)}{2\sigma_\gamma^2}\right) \det(X_\gamma^T X_\gamma + \tau I)^{-1/2}. \tag{141}
$$

By combining (125) and (141), we obtain

$$p(Y|\gamma_i = 1, \gamma_{-i}, X)$$

$$= \int_{\beta_\gamma} \int_{\sigma_\gamma^2=0}^\infty \frac{1}{\left(\sigma_\gamma \sqrt{2\pi}\right)^N} \exp\left(-\frac{\|Y - X_\gamma \beta_\gamma\|^2}{2\sigma_\gamma^2}\right) p(\beta_\gamma|\gamma_i = 1, \gamma_{-i}) p(\sigma_\gamma^2|\gamma_i = 1, \gamma_{-i}) d\beta_\gamma d\sigma_\gamma^2 \tag{142}$$

$$= \int_{\sigma_\gamma^2=0}^\infty \texttt{InvGamma}\left(\frac{1}{2}\nu_0, \frac{1}{2}\nu_0\lambda_0\right) \frac{1}{\left(\sigma_\gamma \sqrt{2\pi}\right)^N} \tau^{|\gamma|/2}$$

$$\times \exp\left(-\frac{(\|Y\|^2 - \|\tilde{Y}_\gamma|^2)}{2\sigma_\gamma^2}\right) \det(X_\gamma^T X_\gamma + \tau I)^{-1/2} d\sigma_\gamma^2 \tag{143}$$

$$= \det(X_\gamma^T X_\gamma + \tau I)^{-1/2} \tau^{|\gamma|/2} (2\pi)^{-N/2} \int_{\sigma_\gamma^2=0}^\infty \texttt{InvGamma}\left(\frac{1}{2}\nu_0, \frac{1}{2}\nu_0\lambda_0\right)(\sigma_\gamma^2)^{-N/2}$$

$$\times \exp\left(-\frac{(\|Y\|^2 - \|\tilde{Y}_\gamma\|^2)}{2\sigma_\gamma^2}\right) d\sigma_\gamma^2 \tag{144}$$

$$= \det(X_\gamma^T X_\gamma + \tau I)^{-1/2} \tau^{|\gamma|/2} (2\pi)^{-N/2}$$

$$\times \int_{\sigma_\gamma^2=0}^\infty \frac{(1/2\lambda_0\nu_0)^{1/2\nu_0}}{\Gamma(1/2\nu_0)} (1/\sigma_\gamma^2)^{1/2\nu_0+1} \exp\left(-1/2\nu_0\lambda_0/\sigma_\gamma^2\right)(\sigma_\gamma^2)^{-N/2}$$

$$\times \exp\left(-\frac{(\|Y\|^2 - \|\tilde{Y}_\gamma\|^2)}{2\sigma_\gamma^2}\right) d\sigma_\gamma^2 \tag{145}$$

$$= \det(X_\gamma^T X_\gamma + \tau I)^{-1/2} \tau^{|\gamma|/2} (2\pi)^{-N/2} \frac{(1/2\lambda_0\nu_0)^{1/2\nu_0}}{\Gamma(1/2\nu_0)}$$

$$\times \int_{\sigma_\gamma^2=0}^\infty (1/\sigma_\gamma^2)^{1/2\nu_0+1+N/2} \exp\left(-\frac{(\|Y\|^2 - \|\tilde{Y}_\gamma\|^2 + \nu_0\lambda_0)}{2\sigma_\gamma^2}\right) d\sigma_\gamma^2 \tag{146}$$

$$= \det(X_\gamma^T X_\gamma + \tau I)^{-1/2} \tau^{|\gamma|/2} (2\pi)^{-N/2} \frac{(1/2\lambda_0\nu_0)^{1/2\nu_0}}{\Gamma(1/2\nu_0)}$$

$$\times \Gamma\left(\frac{N + \nu_0}{2}\right)\left(\frac{\|Y\|^2 - \|\tilde{Y}_\gamma\|^2 + \nu_0\lambda_0}{2}\right)^{-\frac{N+\nu_0}{2}}. \tag{147}$$

Let $\tilde{\gamma}_1$ is given by $\gamma_{-i}$ with $\gamma_i = 1$, $\tilde{\gamma}_0$ is given by $\gamma_{-i}$ with $\gamma_i = 0$. It follows that

$$\frac{p(Y|\gamma_i = 1, \gamma_{-i}, X)}{p(Y|\gamma_i = 0, \gamma_{-i}, X)} = \sqrt{\tau} \sqrt{\frac{\det(X_{\tilde{\gamma}_0}^T X_{\tilde{\gamma}_0} + \tau I)}{\det(X_{\tilde{\gamma}_1}^T X_{\tilde{\gamma}_1} + \tau I)}} \left(\frac{\|Y\|^2 - \|\tilde{Y}_{\tilde{\gamma}_0}\|^2 + \nu_0\lambda_0}{\|Y\|^2 - \|\tilde{Y}_{\tilde{\gamma}_1}\|^2 + \nu_0\lambda_0}\right)^{\frac{N+\nu_0}{2}}. \tag{148}$$

On the other hand, we have

$$\|\tilde{Y}_\gamma\|^2 = \tilde{Y}_\gamma^T \tilde{Y}_\gamma \tag{149}$$

$$= Y^T X_\gamma (X_\gamma^T X_\gamma + \tau I)^{-1} X_\gamma^T Y. \tag{150}$$

Hence, we finally have

$$\frac{p(Y|\gamma_i = 1, \gamma_{-i}, X)}{p(Y|\gamma_i = 0, \gamma_{-i}, X)} = \sqrt{\tau \frac{\det(X_{\tilde{\gamma}_0}^T X_{\tilde{\gamma}_0} + \tau I)}{\det(X_{\tilde{\gamma}_1}^T X_{\tilde{\gamma}_1} + \tau I)}} \left(\frac{S_{\tilde{\gamma}_0}}{S_{\tilde{\gamma}_1}}\right)^{N+\nu_0}, \tag{151}$$

where

$$S_\gamma := Y^T Y - Y^T X_\gamma (X_\gamma^T X_\gamma + \tau I)^{-1} X_\gamma^T Y + \nu_0\lambda_0. \tag{152}$$

Based on this, we can estimate

$$p(\gamma_i|\mathcal{D}, \gamma_{-i}) = \frac{p(\gamma_i|\mathcal{D}, \gamma_{-i})}{p(1 - \gamma_i|\mathcal{D}, \gamma_{-i})}\left(1 + \frac{p(\gamma_i|\mathcal{D}, \gamma_{-i})}{p(1 - \gamma_i|\mathcal{D}, \gamma_{-i})}\right)^{-1}. \tag{153}$$

Denote the set of included variables in $\tilde{\gamma}_0$ as $I = \{j : \tilde{\gamma}_{0,j} = 1\}$. Define $F = \left(X_{\tilde{\gamma}_0}^T X_{\tilde{\gamma}_0} + \tau I\right)^{-1}$, $\nu = X^T Y$ and $\nu_{\tilde{\gamma}_0} = (\nu_j)_{j \in I}$. Also define $A = X^T X$ and $a_i = (A_{ji})_{j \in I}$. Then, by using the same arguments as (Zanella & Roberts, 2019, Appendix B1), we can show that

$$S(\tilde{\gamma}_1) = S(\tilde{\gamma}_0) - d_i\left(\nu_{\tilde{\gamma}_0}^T F a_i - \nu_i\right)^2, \tag{154}$$

where $d_i = (A_{ii} + \tau - a_i^T F a_i)^{-1}$. In addition, we can compute $a_i^T F a_i$ by using the Cholesky decomposition of $F = LL^T$ and

$$a_i^T F a_i = \|a_i^T L\|^2 \tag{155}$$

$$= \sum_{j \in I}(BL)_{ij}^2, \tag{156}$$

where $B$ is the $p \times |\gamma|$ matrix made of the columns of $A$ corresponding to variables included in $\gamma$.

In addition, we have

$$X_{\tilde{\gamma}_1}^T X_{\tilde{\gamma}_1} + \tau I = \begin{pmatrix} X_{\tilde{\gamma}_0}^T X_{\tilde{\gamma}_0} + \tau I & a_i \\ a_i^T & A_{ii} + \tau \end{pmatrix} \tag{157}$$

Hence, by using Schur's formula for the determinant of block matrix, we are easy to see that

$$\frac{\det(X_{\tilde{\gamma}_0}^T X_{\tilde{\gamma}_0} + \tau I)}{\det(X_{\tilde{\gamma}_1}^T X_{\tilde{\gamma}_1} + \tau I)} = d_i. \tag{158}$$

Using this algorithm, if pre-computing $X^T X$ is not possible, the computational complexity per conditional PIP is $O(N|\gamma|^2 + |\gamma|^3 + P|\gamma|^2)$. Otherwise, if pre-computing $X^T X$ is possible, the computational complexity per conditional PIP is $O(|\gamma|^3 + P|\gamma|^2)$.

## F  Proof of Lemma 9

From Lemma 8 and the fact that $\{\gamma^{(t)}, Q^{(t)}\}$ forms a reversible Markov chain with transition kernel $K((\gamma, Q) \to (\gamma', Q')) = K^*(\gamma \to \gamma')q(Q')$, we have

$$1 - \lambda_{\gamma,Q}$$

$$= \inf_{g(\gamma,Q):\mathbb{E}_\pi[g]=0,\mathbb{E}_\pi[g^2]=1} \langle g, g \rangle_\pi - \langle \mathbf{K}g, g \rangle \tag{159}$$

$$= 1 - \sup_{g(\gamma,Q):\mathbb{E}_\pi[g]=0,\mathbb{E}_\pi[g^2]=1} \langle \mathbf{K}g, g \rangle \tag{160}$$

$$= 1 - \sup_{g(\gamma,Q):\mathbb{E}_\pi[g]=0,\mathbb{E}_\pi[g^2]=1} \sum_{\gamma,Q} \mathbf{K}g(\gamma,Q)g(\gamma,Q)\pi(\gamma,Q) \tag{161}$$

$$= 1 - \sup_{g(\gamma,Q):\mathbb{E}_\pi[g]=0,\mathbb{E}_\pi[g^2]=1} \sum_{\gamma,Q}\sum_{\gamma',Q'} K((\gamma,Q) \to (\gamma',Q'))g(\gamma',Q')g(\gamma,Q)\pi(\gamma,Q) \tag{162}$$

$$= 1 - \frac{S}{P} \sup_{g(\gamma,Q):\mathbb{E}_\pi[g]=0,\mathbb{E}_\pi[g^2]=1} \sum_{\gamma,Q}\sum_{\gamma',Q'} K^*(\gamma \to \gamma')q(Q')g(\gamma',Q')g(\gamma,Q)\pi(\gamma,Q) \tag{163}$$

$$= 1 - \frac{S}{P} \sup_{g(\gamma,Q):\mathbb{E}_\pi[g]=0,\mathbb{E}_\pi[g^2]=1} \sum_{\gamma,Q}\sum_{\gamma',Q'} K^*(\gamma \to \gamma')\frac{f(\gamma)}{Z_f}q(Q)g(\gamma',Q')g(\gamma,Q)q(Q') \tag{164}$$

$$= 1 - \frac{S}{P} \sup_{g(\gamma,Q):\mathbb{E}_\pi[g]=0,\mathbb{E}_\pi[g^2]=1} \sum_{\gamma,\gamma'} K^*(\gamma \to \gamma')\frac{f(\gamma)}{Z_f}\sum_{Q,Q'} g(\gamma',Q')g(\gamma,Q)q(Q)q(Q') \tag{165}$$

$$= 1 - \frac{S}{P} \sup_{g(\gamma,Q):\mathbb{E}_\pi[g]=0,\mathbb{E}_\pi[g^2]=1} \sum_{\gamma,\gamma'} K^*(\gamma \to \gamma')\pi(\gamma)\left(\sum_Q g(\gamma,Q)q(Q)\right)\left(\sum_{Q'} \pi(\gamma',Q')q(Q')\right) \tag{166}$$

$$= 1 - \frac{S}{P} \sup_{g(\gamma,Q):\mathbb{E}_\pi[g]=0,\mathbb{E}_\pi[g^2]=1} \sum_{\gamma,\gamma'} K^*(\gamma \to \gamma')\pi(\gamma)h(\gamma)h(\gamma') \tag{167}$$

where

$$\pi(\gamma) = \frac{f(\gamma)}{Z_f}, \tag{168}$$

$$Z_f = \sum_{\gamma} f(\gamma), \tag{169}$$

$$h(\gamma) := \sum_{Q} g(\gamma, Q) q(Q). \tag{170}$$

Observe that

$$\mathbb{E}_{\pi}[h(\gamma)] = \sum_{\gamma} h(\gamma)\pi(\gamma) \tag{171}$$

$$= \sum_{\gamma} \sum_{Q} g(\gamma, Q) q(Q) \pi(\gamma) \tag{172}$$

$$= \sum_{\gamma, Q} g(\gamma, Q) \pi(\gamma, Q) \tag{173}$$

$$= \mathbb{E}_{\pi}[g(\gamma, Q)] \tag{174}$$

$$= 0. \tag{175}$$

On the other hand, we also have

$$\mathbb{E}_{\pi}\left[h^2(\gamma)\right] = \sum_{\gamma} \left(\sum_{Q} g(\gamma, Q) q(Q)\right)^2 \pi(\gamma) \tag{176}$$

$$\leq \sum_{\gamma} \left(\sum_{Q} g(\gamma, Q)^2 q(Q)\right) \pi(\gamma) \tag{177}$$

$$= \sum_{\gamma, Q} g(\gamma, Q)^2 \pi(\gamma, Q) \tag{178}$$

$$= \mathbb{E}_{\pi}\left[g(\gamma, Q)^2\right] \tag{179}$$

$$= 1, \tag{180}$$

where (177) follows from the convexity of the function $x^2$ on $[0, \infty)$.

From (175), (180), and (167), we obtain

$$1 - \lambda_{\gamma, Q} \geq 1 - \sup_{h(\gamma): \mathbb{E}_{\pi}[h] = 0, \mathbb{E}_{\pi}[h^2] \leq 1} \sum_{\gamma, \gamma'} K^*(\gamma \to \gamma') \pi(\gamma) h(\gamma) h(\gamma'). \tag{181}$$

Now, note that $\mathbb{E}_{\pi}[h] = 0$ is equivalent to $h \perp_{\pi} \mathbf{1}$. Let $|\Omega| = 2^{P+1} := n$ and $h_1, h_2, \cdots, h_n$ are eigenfunctions of $\mathbf{K}^*$ corresponding to the decreasing ordered eigenvalues $\lambda_1 \geq \lambda_2 \geq \cdots \geq \lambda_n$ and are orthogonal since $\mathbf{K}^*$ is self-adjoint. Set $h_1 = \mathbf{1}$. Since $\|h\|_{2,\pi} = 1$ and $h \perp_{\pi} \mathbf{1}$, we have $h = \sum_{j=2}^{n} a_j h_j$ because it is perpendicular to $h_1$ so it can be only represented by these eigenvectors. By taking $l_2$-norm on both sizes we have $\sum_{j=2}^{n} a_j^2 \leq 1$ since the form like $\langle h_i, h_j \rangle_{\pi} = 0$ and $\langle h_i, h_i \rangle = \|h_i\|_{2,\pi}^2 = 1$. Thus,

$$\sup_{h: \mathbb{E}_{\pi}[h] = 0, \mathbb{E}_{\pi}[h^2] \leq 1} \sum_{\gamma, \gamma'} K^*(\gamma \to \gamma') \pi(\gamma) h(\gamma) h(\gamma') \leq \max_{a_2, a_3, \cdots, a_n: \sum_{j=2}^{n} a_j^2 \leq 1} \sum_{j=1}^{n} a_j^2 \lambda_j \tag{182}$$

$$\leq \lambda_2 \sum_{j=2}^{n} a_j^2 \tag{183}$$

$$= \lambda_2, \tag{184}$$

where $\sum_{j=2}^{n} a_j^2 \leq 1$ and $\lambda_j \in \texttt{spec}(P)$ such that $\lambda_2 \geq \lambda_3 \cdots \geq \lambda_n$. Hence, from (184), we obtain

$$1 - \lambda_{\gamma,Q} \geq 1 - \frac{S}{P}\lambda_2 \tag{185}$$

$$= \frac{S}{P}(1 - \lambda_P) + 1 - \frac{S}{P} \tag{186}$$

$$\geq 1 - \frac{S}{P}. \tag{187}$$

