# OpenReview forum: "Variable Complexity Weighted-Tempered Gibbs Samplers for Bayesian Variable Selection"
_TMLR — Withdrawn by Authors_

### Review · Reviewer_idXd · 2024-03-04

**Summary Of Contributions:**

The authors propose a variant of the subset weighted-tempered Gibbs Sampler (subset-wTGS) introduced by (Jankowiak, 2022), with the goal of reducing the computational cost per iteration compared to the original alternative.

Because generating random sets of size $S$ from $P$ choose $S$ subsets is computationally demanding in the original algorithm, the authors avoid such computations at every interaction. This is achieved by leveraging a stochastic decision process, driven by a Bernoulli distribution with probability $S/P$, of whether to update the Gibbs posteriors of interest or not.

Unlike traditional subset-wTGS and wTGS, the proposed algorithm is a stochastic version of the original one, where the number of posterior inclusion probability (PIP) computations per-interaction is now random, i.e., it exhibits variable complexity.

The authors showcase that

- the average number of conditional PIP computations is $S = P \times (S/P)+ 0 \times (1−S/P)$, i.e., it is much smaller than the problem dimension.
- the newly proposed estimator converges to the PIPs of interest.
- the variance of the proposed Rao-Blackwellized estimator scales quadratically with the ratio of P to S, where S represents the average number of conditional PIP computations per iteration. Hence, the variance of the Rao-Blackwellized estimator can be much smaller than the subset wTGS proposed by (Jankowiak, 2022), for high $P/S$ ratios.

**Audience:**

Yes

**Claims And Evidence:**

No

**Requested Changes:**

I would request that the authors address the following questions to improve the significance of their contribution

1. Definition, clarity and implications of the studied problem
    - The authors describe in the introduction the subset wTGS algorithm as a general MCMC approach.
    - However, in their problem set-up of section 2.2, they specify and focus on the Bayesian Variable selection problem for the linear regression case.
    - Hence,
        - I would encourage the authors to specify the problem they are targeting as early as possible.
        - Besides, an earlier mathematical definition of PIP, now defined in Equation (9) of section 2.2, would help clarify the metric of interest.
    - In addition, can the authors clarify the following questions?
        - How important are the linear regression model assumptions (i.e., linearity and Gaussian noise) for their proposed algorithm?
        - How important are the linear regression model assumptions (i.e., linearity and Gaussian noise) in their proofs? I.e., in which proof steps are these assumptions critical?
        - Is it possible (and if so, what are the steps) to apply or extend VC-wTGS to non-Gaussian and-or non-linear regression models?

1. Notation and clarity regarding PIP, $i$ and PIP($i$):
    - In equation (9), PIP is defined in terms of each feature index $i=1, 2, \cdots, P$, as the conditional probability of each $i$-th covariate being included. Namely, $i$ is an index for each covariate, and $PIP(i)$ the corresponding posterior probability of index $i$ being included in the regression.
    - In Equation (10), $i$ appears on the left hand side of the equation, as a random variable. The authors, within the $i$-updates of Section 3.1, refer to (a) the marginalization of $i$ from (10), and (b) computing a conditional distribution over $i$ in Equation (13). Can the authors explain whether these $i$ are random or deterministic, and how do they relate to the index $i=1, 2, \cdots, P$ in Equation (10)?
    - In the experimental section, the authors' refer to the "PIP-estimator", the "value of PIP", "PIP(0)" and "PIP(1)" (as illustrated in Figures 1 and 2). According to the definition in equation (9) and the proposed estimator in Equation (15), there are $P$ PIP quantities to be estimated, one for each $i \in P$. What exactly are the authors evaluating in these figures?

1. Computation of different subset sizes $S_t$ per interaction
     - In Section 3.2, the authors argue they "propose a variable complexity-based wTGS scheme", where they use different subset sizes $S_t$ at each MCMC iteration (under the condition that the expected number of the conditional PIP computations per MCMC iterations is equal to the a priori defined subset size S).
     - However, inspection of Algorithm 2 implies that
        - one computes $P$ conditional PIPs with probability $S/P$, or
        - there are no computations made, with probability $1-S/P$.
    - Hence, although it is true that the average number of conditional PIP computations is $S = P \times (S/P)+ 0 \times (1−S/P)$, it seems that only two possible number of subset sizes $S_t$ can be computed at each interaction ($P$ or $0$). Please clarify whether this is the case.
    - It would be of interest to demonstrate the evolution of the number of computations per MCMC interaction over interactions $T$ of Algorithm 2. I.e., plots showcasing the number of computations per MCMC interaction in their experiments would better convey their claim about the "variable complexity" nature of Algorithm 2, and that presented "plots show a trade-off between the computational complexity and the estimated variance for estimating PIP".

1. Subset size S as a hyperparameter.
    - The authors argue that the proposed method provides lower variance when $P/S > 1$, as per the presented upper-bound and the provided empirical evidence.
    - Can they please elaborate on whether the $P/S >1$ case is the only case of interest? i.e., is this work significant only in the sparse Bayesian regression setting? Clarifications would be greatly appreciated.
    - Additionally, can the authors elaborate on how does one decide what $S$ to use in practice?

1. Experimental results
    - The meaning of the Y-axis in all figures (i.e., PIP(0) and PIP(2)) should be clarified ---see previous notation questions on $i$ and $PIP(i)$
    - Figure 1 showcases the convergence of the proposed algorithm to ground truth (in simulated data). However, the convergence of the original subset wTGS by (Jankowiak, 2022) is not evaluated in the simulated scenario of Figure 1. A comparison would strengthen the significance of this experiment, given that Jankowiak (2022) is the most similar algorithm to the proposed one.
    - I would encourage the authors to provide computational complexity plots (i.e., number of computations per-MCMC interaction) for both algorithms in these simulated examples.
    - In Section 4.2., the authors claim that "The expected number of PIP computations is only $ST$ in proposed ALG 2, but $TP$ in wTGS if we run T MCMC iterations", which is correct. However, the authors' acknowledge that "In the subset wTGS in Subsection 3.1, the number of conditional PIP computations per MCMC iteration is fixed, i.e., it is equal to S$", which, to the best of my knowledge, is the algorithm they are comparing to in Figures 2 and 3 (please correct me if I am wrong). The provided plots show the variance of wTGS and VC-wTGS results. Can the authors clarify what are the insights learned from Figures 2 and 3 with respect to computational complexity of these algorithms?
    - In addition, the authors claim that Fig. 3 shows that VC-wTGS outperforms subset wTGS (Jankowiak, 2022) at high values of $P/S$. The contribution would be strengthen by providing a comparison for smaller values of $P/S$ as well, e.g., $P/S \approx 1$.
    - Finally, I would appreciate it if the authors could clarify what is the motivation and modeling assumptions for evaluating these algorithms in the MNIST dataset, instead of more simulated scenarios where access to ground truth is possible.

1. Clarity and and preciseness suggestions
    - The following reference for particle filters (Bugallo et al., 2007) in the first paragraph of the introduction seems quite limited. A more general, introduction to PF reference seems more suitable.
    - Second paragraph of introduction, "obtaining a sequence of observations which are approximated from a specific multivariate probability distribution" does not seem correct, as a Gibbs sampler obtains samples from a distribution, not observations that are approximations.
    - Third paragraph of introduction, there are references to "important sampling" when I believe the authors are referring to "importance sampling".
    - The definition of $P$ in the fourth paragraph of the introduction, "the signal dimension, P" is presented without clarifying what the authors mean by signal in the context of Bayesian variable selection. I would encourage to clarify so, and also suggest to clearly define variable $S$ in this paragraph, before referring to the ratio $P/S$.
    - In the fourth paragraph of the introduction, unclear sentence "that results whenever there more than a few dozen covariates"
    - Other minor typos, e.g., "$\eta(\cdot)$ is some weighting functions" after equation 10 in Section 3.1
    - Section 3.2: "propose a variable complexity-based wTGS schemes"

**Strengths And Weaknesses:**

Strengths:

- Bayesian Variable Selection is a relatively important problem, for which MCMC techniques are often used, and of interest to many practitioners.
- The authors demonstrate that the proposed stochastic version of the w-STG algorithm has an average of $S$ number of computations per interaction
- The authors provide important theoretical results in Theorem 9, where they show convergnece of the proposed estimator and its variance is upper bounded, with a quadratic dependency on the P to S ratio.

Weaknesses:
- The proposed method's contribution is limited to reducing the average computational cost per interaction.
- The reduced variance of the proposed estimator occurs only for high values of the $P/S$ ratio.
- The clarity of the manuscript can be improved.
- The experiment section can be improved for a more clear assessment of the benefits of the proposed algorithm.

---

### Review · Reviewer_esQU · 2024-03-11

**Summary Of Contributions:**

This paper proposes an algorithm (weighted tempered Gibbs sampler) for Bayesian variable selection. The authors provide theoretical guarantees by offering an upper bound on the variance of an estimator for the posterior inclusion probabilities (PIPs), as well as a lower bound on the spectral gap of the sampler.

**Audience:**

Yes

**Broader Impact Concerns:**

None.

**Claims And Evidence:**

Yes

**Requested Changes:**

In section 2.2, the notation is unclear: does $\beta_\gamma$ belong to $\mathbb R^P$ or to $\mathbb R^{|\gamma|}$? I suggest always writing $\beta \in \mathbb R^P$ and writing $\beta_\gamma$ to denote the restriction of $\beta$ to the coordinates in $\gamma$.

Below eq. (13), it is claimed that the marginal $f(i)$ is proportional to $\texttt{PIP}(i) + \varepsilon/P$; please justify.

Algorithm 1: what is $\phi_0$?

In Lemma 5, how small can $\min_\gamma \pi(\gamma)$ be? Does this lead to a poor quantitative dependence for the final bound?

Could you include a more detailed discussion on how to compute the conditional PIPs in general?

Typos:
- Pg. 2, "has very high variance as $P/S$ large at a small number of MCMC iterations, $T$" is a sentence fragment
- Below eq. (5): $Q$ is not the infinitesimal generator, $I-Q$ is
- Eq. (8): in the numerator, it should be $\nu$ not $v$
- Below eq. (8): what is meant by the "standard $L_2$ norm"? The state space $\mathcal S$ does not come equipped with a measure in this section (unless you restrict to $\mathcal S$ with finite cardinality and take the uniform measure?)
- Below eq. (10): "if size $A < S$" -> "of size $A < S$"; "some weighting functions" -> "some weighting function"

**Strengths And Weaknesses:**

Strengths:

The proposed algorithm is novel and comes with strong theoretical guarantees. Bayesian variable selection is a practically relevant problem that should be of interest to the TMLR community.

Weaknesses:

Overall, I found the paper to not be clearly written. For example, it could benefit from a more detailed discussion on the practical implementation of the sampler, particularly regarding the computation of conditional PIPs in more complex models. Also, the paper relies heavily on the previously proposed TGS algorithm, but since it does not describe it in enough detail, it makes it difficult to understand the newly proposed algorithm.

---

### Review · Reviewer_rj4i · 2024-03-18

**Summary Of Contributions:**

This paper considers a variable selection problem for an additive noise linear model.
Authors specifically consider a Bayesian approach where the inclusion of a covariate is expressed by a Berounlli random variable -- the goal is to compute the posterior probability of including a covariate given observed data (termed posterior inclusion probabilities (PIPs)).

The paper proposes an MCMC sampler which serves as an extension to the weighted tempered Gibbs sampler (wTGS) proposed by Jankowiak, 2022.
The proposed algorithm essentially follows the wTGS with an additional random variable that decides to conduct a (possibly) costly Gibbs update.
Authors show the proposed approach provide a consistent estimate of the PIP and derives a variance upper bound.

**Audience:**

Yes

**Claims And Evidence:**

No

**Requested Changes:**

Please address the weaknesses above

**Strengths And Weaknesses:**

## Strengths

* The consistency and convergence rate analysis (variance) could be useful.
* The proposed sampler seems to be a faster approach than that of Jankowiak, 2022.

## Weaknesses

Experiment descriptions lack details required for replication and assessing the proposed method:

* In both synthetic and real data experiments, the design of the PIPs are not given (only the details of covariates are given).
  * The convergence reported in Fig 1 (synthetic experiment) does not make sense, since the ground-truth is not given to the reader.
  * There does not seem to be a well-defined notion of a PIP for a real-world dataset unless we create a dataset following the generative model. In particular, how are the "estimated variance" computed?.
* The comparison to the subset wTGS (baseline) is limited to a single setting. A more extensive evaluation is desirable, such as comparison in terms of computing time and different values of $P/S$.

Apart from the issues with the experiment section, the paper is overall not well-written. Some comments:
1. The first two paragraphs of the introduction seems to be too generic. The paper's scope appears to be very narrow.
2. It is unclear how the background information in Section 2 is used in the rest of the paper.
3. The introduction to the subset wTGS is not really helpful. The motivation for using the target distribution (Eq. 10) is not clear until the invariant distribution is given in Lemma 3.
4. Algorithm 1: "Estimate $f(j|\gamma^{(t-1)})$". Is this only an estimate?
5. The same symbol $\mathrm{PIP}(i)$ is used for different estimators.
6. I don't really follow the argument "generating a random set $\mathcal{S}$ from  $P \choose S$subset. Since the anchor set narrows down the possible sets, the computation might not be that expensive?
7.  I am not sure what part is "Rao-Blackwellised" in the proposed estimator (15).
8. The presentation of the theoretical results in Section 3.3 could be improved.
  * In general it is helpful to outline the results to be presented at the outset.
  * There are technical lemmas that are not directly used in the results presented in the main text (e.g., Lemma 1 and 2).
  * Separating the main result (Theorem 9?) and the proofs would be helpful. The current structure requires readers to navigate through technical lemmas before reaching the main result. A clearer presentation with the result upfront would enhance readability.
9. In my understanding, $\min_{\gamma} \pi(\gamma$ could be really small in a sparse setting, and thus the bound in Theorem 9 might have a really large constant.

---

### Note · Authors · 2024-03-19

I have read and agree with the venue's withdrawal policy on behalf of myself and my co-authors.